# Impact of Assimilated Observations on the Corsica Channel Transport in a 4D-Var System for the Northwestern Mediterranean Sea

Michele Bendoni[1], Andrew Michael Moore[2], Roberta Sciascia[1], Carlo Brandini[3], Katrin Schroeder[4], Mireno Borghini[1], and Marcello Gatimu Magaldi[1]

[1]CNR-ISMAR, 19032 Lerici (Sp), Italy
[2]Department of Ocean Sciences, University of California, 95062 Santa Cruz CA
[3]CNR-ISMAR, 50019 Sesto Fiorentino (Fi), Italy
[4]CNR-ISMAR, 30122 Venezia (Ve), Italy

**Correspondence:** Michele Bendoni (michele.bendoni@cnr.it)

**Abstract.** We present a 4D-Var data assimilation (DA) system covering the North-Western Mediterranean Sea implemented with the Regional Ocean Modeling System (ROMS). We study, throughout the year 2022, its ability to improve the description of the overall circulation and the capability to constrain the transport across the Corsica Channel (CC), the dynamics of which are crucial in determining the circulation throughout the region. The system assimilates Sea Surface Temperature (SST) and Sea Level Anomaly (SLA) observations from satellites, surface velocity data from High-Frequency Radars (HFRs), and in situ temperature, salinity and velocity observations, the latter from a mooring located in the CC. For all the observed state variables, DA is able to improve the forecast and the analysis compared to a free run without DA, with root mean squared error reduction up to 60% and correlation increase up to 0.4. The general circulation after DA is characterized by a reduction of the Eastern Corsica Current (ECC) and an increase of the Western Corsica Current (WCC). An adjoint sensitivity-based method was used to evaluate the impact of observations on the CC transport state estimates. The net reduction in transport induced by DA, changed the annual average value of 0.49 Sv for the free run, to 0.31 Sv and 0.28 Sv for the forecast and analysis, respectively. The observations that contribute most to the transport changes are the in situ velocity data and those from HFRs. The observation impacts were found to vary seasonally, and sometimes act in competition to shape the circulation pathways across the CC. The sensitivity of the transport to SST and in situ temperature and salinity observations indicates that remote measurements (e.g. those from the Gulf of Lion) can potentially play a significant role in constraining the CC transport. Transport variations are largely affected by free surface gradients contained in the increment, and promoted by modifications to the open boundary conditions. This indicates that the CC dynamic is controlled by mechanisms operating at basin scales.

## 1 Introduction

Data assimilation (DA) is essential for ensuring the quality of both forecasts and reanalysis, in the atmosphere as well as in the ocean. Several ocean models implement 4D-Variational Data Assimilation (4D-Var DA) systems for practical applications in various regions of the global ocean, where observational coverage is sufficient to meaningfully constrain the circulation

estimates. For example 4D-Var DA system have been implemented at the global ocean scale by the Japan Meteorological Agency (Fujii et al., 2023), and at regional scales in the area between Australia and Indonesia (Janeković et al., 2022), in the coastal waters of eastern Australia (Zavala-Garay et al., 2012; Kerry et al., 2020), in the coastal waters of North America (Zhang et al., 2010; Moore et al., 2013), around several island nations of the Pacific (Arango et al., 2011; Janeković et al., 2013), in the coasts of North Europe (Sperrevik et al., 2015), and in the Mediterranean Sea with specific application in the coastal area of the Tyrrhenian Sea (Iermano et al., 2016), in the Adriatic Sea (Janeković et al., 2020), and in north-western part of it (Bendoni et al., 2023).

The north-western Mediterranean Sea, also known as Liguro-provençal Basin, is a crucial maritime region, hosting the Pelagos Sanctuary (Notarbartolo di Sciara et al., 2008) and several Marine Protected Areas (Francour et al., 2001). It is characterized by a cyclonic circulation (Northern Gyre) that involves Atlantic Water (AW) and modified Atlantic Water (mAW) at the surface, and Eastern Intermediate Water (EIW) and Tyrrhenian Intermediate Water (TIW) at intermediate depths. Winter Intermediate Water (WIW) is mostly located at intermediate depth in the western part of the basin, whereas Western Mediterranean Deep Water (WMDW) occupy the deeper layers (Astraldi et al., 1994; Napolitano et al., 2019; Schroeder et al., 2024; Barral et al., 2024).

Around Corsica, the Eastern Corsica Current (ECC) and the Western Corsica Current (WCC) flow along opposite coasts and converge to form the Northern Current (NC, or Liguro-Provençal-Catalan Current), which flows cyclonically along the Italian and French coasts, up to the Catalan coast (Astraldi et al., 1990; Millot, 1999). The ECC passes through the Corsica Channel (CC), a narrow strait between northern Corsica and the Capraia Island, about 30 km wide at the surface and 450 m deep. This strait represents the main connection between the warmer, saltier Tyrrhenian waters and the colder, fresher waters of the Liguro-Provençal basin (Bethoux, 1980; Astraldi et al., 1990).

The northward transport across the CC shows a marked seasonal cycle, with higher values during winter and spring and a net reduction in summer and autumn (Astraldi and Gasparini, 1992), occasionally reversing direction (Sciascia et al., 2019). This modulation aligns to the seasonal dynamics of the Tyrrhenian Sea. In winter, it is characterized by a large-scale cyclonic circulation, when both surface and intermediate waters flow along the Italian coast and bifurcate, one part reaching the CC, and the other veering southward to join a semi-permanent cyclonic structure close to the Bonifacio Strait called Bonifacio Gyre. In summer, most of the water mass is recirculated within the Tyrrhenian basin with little outflow toward the Ligurian Sea (Astraldi and Gasparini, 1994; Artale et al., 1994). In addition, the modulation of transport through the CC can be affected by the presence of a recurrent anticyclonic structure, peculiar of the summer season and located in the channel area, known as Ligurian Anticyclone (LA) (Ciuffardi et al., 2016; Iacono and Napolitano, 2020).

The CC significantly influences Western Mediterranean dynamics by affecting the Ligurian Current and the WIW and WMDW formation process (Schroeder et al., 2010), shaping biological connectivity between the Tyrrhenian and Ligurian Seas (Aliani and Meloni, 1999) depending on flow strength (Astraldi et al., 1995), and playing a key role in the accumulation and dispersal of floating debris and pollutants (Fossi et al., 2017). This makes it a critical area for understanding oceanographic, ecological, and environmental processes in the region.

Several DA systems have been implemented at the basin scale in the Mediterranean Sea, targeting both physical (Dobricic et al., 2005; Escudier et al., 2021; Bajo et al., 2023), and biogeochemical variables (Teruzzi et al., 2018). Regional DA applications in the western basin differ by algorithm (e.g., variational methods, ensemble-based Kalman filters) and by scientific objectives. Particular attention has been paid to the assimilation of surface velocities from High-Frequency Radars (HFRs), either alone or in combination with standard observations such as sea surface temperature (SST), sea level anomaly (SLA), and in situ temperature and salinity. Marmain et al. (2014) used HFR observations to adjust atmospheric forcing off the French coast in front of Toulon, and Vandenbulcke et al. (2017) tried to correct the inertial oscillations in the Ligurian Sea. Iermano et al. (2016) and Bendoni et al. (2023) both applied 4D-Var to improve circulation estimates in the Tyrrhenian and north-western Mediterranean, respectively, analyzing the effect of different observation types on alongshore coastal transport, while Hernandez-Lasheras et al. (2021) used HFR data to better resolve mesoscale dynamics in the westernmost part of the Mediterranean basin. In situ data were employed by Hernandez-Lasheras and Mourre (2018) to compare model results after assimilating glider and CTD (conductivity-temperature-depth) observations, and by Aydogdu et al. (2025) to examine the impact of the assimilation of autonomous glider data comparing outputs from three different models with partially overlapping domains. In situ observations from specific datasets, such as CORA (Szekely et al., 2025), have been successfully assimilated into global and regional reanalysis (Jean-Michel et al., 2021; Zuo et al., 2018). Moreover, the assimilation of velocity data from mooring has proven to be a key ingredient to improve the transport, as shown by Panteleev et al. (2016) with a reanalysis of the Eastern Bering Sea circulation using a two-way nested 4D-Var system.

Several modeling studies without DA (Béranger et al., 2005; Sciascia et al., 2019; Poulain et al., 2020) analyzed the dynamic of the CC. Those, instead, using DA (Vandenbulcke et al., 2017; Escudier et al., 2021) and including the CC in their computational domains lack a specific assessment of how assimilation improves the current representation in the channel. Furthermore, starting from the theoretical basis and the numerical algorithms employed to calculate the increment, DA systems offer tools to analyze several aspects of the assimilation procedure. For example, observation impact experiments allow us to evaluate how different data sources affect scalar quantities such as net volume and energy transport across a section, or upwelling intensity (Moore et al., 2011b; Levin et al., 2020, 2021). Evaluating the contribution of different observations to transport increments through the CC helps clarify how DA constrains the model through the relevant physical mechanisms. Furthermore, the spatial distribution of the assimilated observations to which the transport increment is most sensitive reveals the regions that are potentially most influential for transport variability.

In this paper, we present an improved version of the 4D-Var DA system previously implemented by Bendoni et al. (2023) for the north-western Mediterranean, using the ROMS model (Shchepetkin and McWilliams, 2003, 2005). In addition to HFR-derived surface velocities and satellite SST, the system also assimilates SLA, in situ temperature, salinity and current profiles from the mooring system in the Corsica Channel for the year 2022. It also represents an improvement over the Mediterranean Sea Physics and Reanalysis (Escudier et al., 2021), provided by the Copernicus Marine Environment Monitoring Service (CMEMS), by increasing the resolution from 1/24° to 1/36°, the use of a 4D-Var instead of a 3D-Var algorithm, and the assimilation of velocity observations.

 The DA configuration and the observation impact methodology are described in Section 2. The effect of DA on circulation with respect to a non-assimilative model run is addressed in Section 3, focusing both on the agreement with observations (Subsection 3.1), and on the result of the observation impact methodology applied to the Corsica Channel transport (Section 3.2). The discussion about the changes induced by DA on the system and the related mechanisms are reported in Section 4.1 and 4.2, respectively. The last section is dedicated to conclusions and outlooks.

## 2 ROMS 4D-Var DA system for the NWM

### 2.1 Model setup

The ROMS 4D-Var Data Assimilation system (Moore et al., 2011c) implemented for this study (version 4.3), builds upon the configuration developed by Bendoni et al. (2023), with modifications applied to the data assimilation framework, expanding the amount and type of assimilated observations (see Section 2.2) and by extending the analysis to a whole year. Specifically, the same numerical grid (Figure 1a, 1/36° resolution with 40 terrain-following sigma layers) is used and the following model parameters are adopted for the characterization of the vertical discretization of the grid: $V_{transform} = 2$, $V_{stretching} = 4$, $\theta_s = 7$, $\theta_b = 2$, and for the horizontal eddy viscosity $\nu = 5$ m$^2$/s, and diffusivity $\kappa_T = 1$ m$^2$/s. The $k$-$kl$ GLS (Generic Length Scale) scheme for turbulence closure parameters is adopted (Warner et al., 2005). Since the tidal signal in the area is of the order of centimeters, and considering that we use daily averaged values as boundary conditions, we did not take into account tides in the modelling system.

Initial and boundary conditions are derived from the Mediterranean Sea Physics Reanalysis (Escudier et al., 2021) provided by the Copernicus Marine Environment Monitoring Service (CMEMS), at 1/24° resolution and daily frequency. These are imposed at the southern open boundary. Radiation boundary conditions are applied at all depths to velocity components ($u$, $v$), temperature ($T$) and salinity($S$), with a one-day nudging timescale for inflow, three times larger than outflow (Marchesiello et al., 2001). A Flather condition (Flather, 1976) is used for barotropic velocities, a Chapman condition (Chapman, 1985) for the free surface $\eta$, and a zero gradient condition for total kinetic energy (TKE). Atmospheric forcing is provided at hourly frequency from the ERA5 reanalysis dataset (Hersbach et al., 2023), including sea-level pressure, 10-m air temperature and relative humidity, precipitation rate, surface wind components, and downward short- and long-wave radiations. These variables are used to compute air-sea fluxes following the bulk formulation of Fairall et al. (1996). Daily river discharge data for the Rhone and Arno rivers were obtained from the Hydro Portal of France Waters (https://www.hydro.eaufrance.fr), and the Tuscany Region SIR Dataset (https://www.sir.toscana.it), respectively. The nonlinear model without assimilation (freerun, FR) was run from 2019 to 2022, after a three-month spin-up starting in late 2018. The DA experiment spans the period January-December 2022, starting from an initial first-guess provided by the free nonlinear model run on 1 January 2022 at 00:00.

The ROMS state vector comprising all ocean gridpoint values of temperature, salinity, horizontal velocity and free surface height will be denoted by **x**. Since we adopt a strong constraint approach, the control vector **z** includes initial condition, open boundary conditions and surface fluxes but no model error is explicitly considered (Moore et al., 2011c). The analysis vector

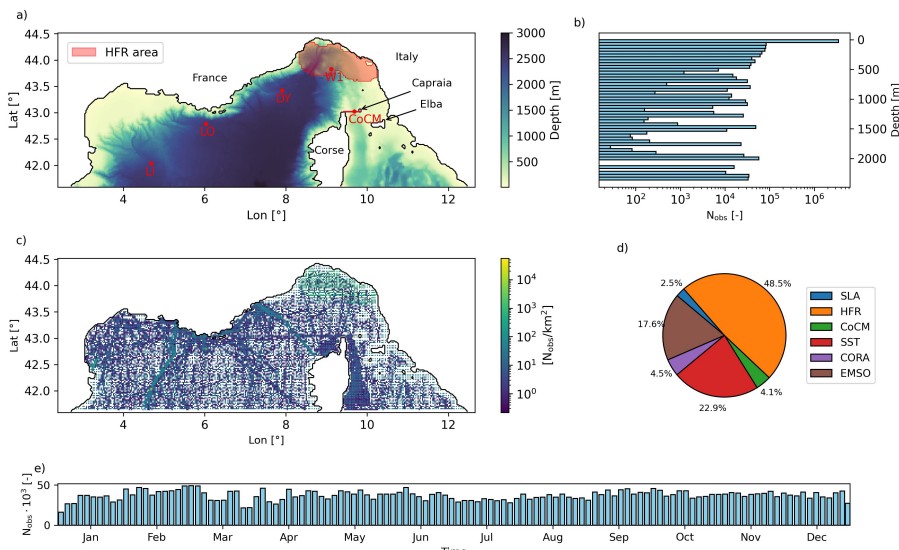

**Figure 1.** a) Model domain and bathymetry. Red dots represent the mooring stations from EMSO (LI: Lion, LO: Albatross, DY: Dyfamed, W1: W1M3A) and JERICO (CoCM: Corsica Channel Mooring). Red shaded area represents the extend of the JERICO HF radar network. The dark red line indicates the Corsica Channel transect to which we refer throughout the paper. b) Vertical distribution with depth of the assimilated observations. c) Spatial horizontal density of the assimilated observations. d) Proportion of the different sources of observations with respect to the total. e) Time evolution of the total number of observations per assimilation window throughout the year 2022.

$\mathbf{z}_a$ in 4D-Var DA can then be expressed by the following equation:

$$\mathbf{z}_a = \mathbf{z}_b + \mathbf{B}\mathbf{G}^T(\mathbf{G}\mathbf{B}\mathbf{G}^T + \mathbf{R})^{-1}\mathbf{d} = \mathbf{z}_b + \mathbf{K}\mathbf{d} \tag{1}$$

where $\mathbf{K}$ is the so called Kalman gain matrix, $\mathbf{z}_b$ is the background control vector, $\mathbf{B}$ is the background error covariance matrix (incorporating background errors associated with initial and boundary conditions and atmospheric forcing), $\mathbf{G}$ is the tangent linear model sampled at observation locations and $\mathbf{G}^T$ its adjoint. $\mathbf{R}$ is the observation error covariance matrix, and $\mathbf{d} = \mathbf{y} - H(\mathbf{x}_b)$ is the innovation vector, with $\mathbf{y}$ the observation vector, $\mathbf{x}_b$ the background state vector, and $H$ the observation operator, which maps from the state space to the observation space and includes the nonlinear model in 4D-Var DA. The matrix $\mathbf{P} = \mathbf{G}\mathbf{B}\mathbf{G}^T + \mathbf{R}$ is the stabilized representer matrix describing the total error covariance in observation space (Moore et al., 2018, 2021).

In practice, the analysis is obtained by minimizing a cost function following the incremental approach of Courtier et al. (1994), as implemented in ROMS by Moore et al. (2011c). Sequential linearizations of the cost function around a nonlinear solution (an "outer loop") similar to a Gauss-Newton method, give rise to a series of quadratic problems which are minimized using the Lanczos formulation of the Restricted $\mathbf{B}$-Preconditioned Conjugate Gradient algorithm (RPCG), implemented as a sequence of "inner loops", as described by Gürol et al. (2014). The algorithm factorizes the $\mathbf{R}$-preconditioned stabilized

representer matrix $\tilde{\mathbf{P}} = \mathbf{R}^{-1}\mathbf{GBG}^T + \mathbf{I}$, yielding an approximation of the Kalman gain matrix of the form:

$$\mathbf{K} \approx \mathbf{BG}^T\mathbf{V}_m\mathbf{T}_m^{-1}\mathbf{V}_m^T\mathbf{GBG}^T\mathbf{R}^{-1} \tag{2}$$

where $\mathbf{V}_m$ is the matrix of Lanczos vectors, $\mathbf{T}_m$ is a tridiagonal matrix, and $m$ is the number of inner loops.

In this study, we use a single outer loop and 9 inner loops trying to find a compromise between the available computational power, the time required to run the experiments and the reduction of the cost function. The analysis/forecast sequence is characterized by a 3-day long assimilation window followed by a 3-day forecast, resulting in 122 windows over the year 2022. No overlapping between analysis (AN) is performed and each analysis uses the forecast (FC) from the previous window as background; hence, in the following, the terms "forecast" and "background" are used interchangeably. The choice of 2022 is motivated by the broader availability of HFR observations during this year.

The background error covariance matrix is modeled following Weaver and Courtier (2001). Climatological variances of model state variables, computed from the free run over 2019-2022, are used as proxies for background error variances. To avoid too large standard deviation values for temperature and heat flux, we removed from the original time series the moving average at quarterly timescale, and then the moving average at daily timescale for each cell of the domain. The background error covariance matrix is factorized as $\mathbf{B} = \boldsymbol{\Sigma}\mathbf{C}\boldsymbol{\Sigma}$ where $\boldsymbol{\Sigma}$ is a diagonal matrix of standard deviations, and $\mathbf{C}$ a univariate correlation matrix with a specified decorrelation length, modeled with a diffusion operator. Horizontal and vertical decorrelation length scales were set to 15 km and 10 m, respectively, for all state variables associated with initial and boundary conditions, and to 70 km for surface fluxes. Although $\mathbf{B}$ is assumed univariate, flow-dependent cross-covariances emerge during the assimilation window via $\mathbf{GBG}^T$ in Equation 1. For practical reasons, the observation error covariance matrix $\mathbf{R}$ is considered diagonal. Error values for each observation type are discussed in section 2.2).

## 2.2 Assimilated observations

Several observational datasets were assimilated into the model, including surface currents from the Tyrrenian-Ligurian HFR network, in situ velocities, temperature and salinity measurements from various platforms, and satellite-derived SST and SLA.

HFR total surface currents $u$ and $v$ from the European HFR Node (https://www.hfrnode.eu) cover parts of the Ligurian Sea (Figure 1a) being JERICO (Joint panEuropean Research Infrastructure for Coastal Observatories, https://www.jerico-ri.eu) facilities. These data are provided at 2.0 km resolution and hourly frequency for 2022. Before assimilation, a 3-hour low-pass filter was applied at each location to the $u$ and $v$ time series to remove high-frequency variability. We did not perform a specific procedure to remove the tidal signal since we assumed the tidally induced velocities to be negligible. Only observations flagged as "good" with a geometric dilution of precision (GDOP) larger than 2 were retained (Fang et al., 2015). Furthermore, at each time step, outliers-defined as velocities above the 99.5%-ile-were excluded. To reduce the HFR data volume and redundancy (Figure 1d), the $u$ and $v$ velocities were averaged into 6 km bins and sub-sampled every 3 hours. Visual inspection confirmed that the spatially averaged fields retained key patterns. Despite this thinning, HFR data still comprised nearly half of all assimilated observations (Figure 1d). The HFR observation error standard deviation was set to 0.1 m/s Hernandez-Lasheras et al. (2021); Bendoni et al. (2023).

In situ velocities were also obtained from the Corsica Channel Mooring (CoCM, Figure 1a) (Aracri et al., 2016), which is also a JERICO facility and includes an upward-facing acoustic doppler current profiler (ADCP) deployed at 450 m depth. The instrument recorded data at 16 m vertical intervals (from 379 m upward) every 2 hours. Observations with a multibeam-derived "indicative error" (GLGOPV) outside the $[-0.1, 0.1]$ m/s range-mostly from depths shallower than 50 m-were excluded due to unrealistic signal shifts. A 6-hour moving average was applied to the time series before assimilation, and the observation error standard deviation was set to 0.05 m/s.

In situ temperature and salinity data were obtained from CMEMS and the European Multidisciplinary Seafloor and water column Observatory (EMSO-ERIC https://emso.eu). CMEMS data (product INSITU_GLO_PHY_TS_DISCRETE_MY_013_001) include time series and profile datasets. They are part of the Global Ocean CORA In situ Observations (Szekely et al., 2025; Cabanes et al., 2013), collected mainly by Argo Floats, XBT, CTD, XCTD and moorings, and we refer to them as $T_{\text{CORA}}$ and $S_{\text{CORA}}$ to indicate temperature and salinity data, respectively, without making the same distinction unless otherwise noted. Only data with all quality flags equal to "good" were used. EMSO data are temperature ($T_{\text{EMSO}}$) and salinity ($S_{\text{EMSO}}$) time series from four different mooring platforms: Lion (LI) in the Gulf of Lion (https://www.seanoe.org/data/00333/44411, Bosse et al. (2025); Houpert et al. (2016)), Albatross (LO) near Toulon (https://www.seanoe.org/data/00720/83244, Lefevre et al. (2016)), Dyfamed (DY) in the western Ligurian Sea (https://www.seanoe.org/data/00326/43749, Coppola et al. (2025)), and W1M3A (W1) in the central Ligurian Sea (www.w1m3a.cnr.it). Locations are reported in Figure 1a. Observation error standard deviations were set to 0.2°C for temperature and 0.075 for salinity.

Satellite-derived SST was obtained from the CMEMS Mediterranean Sea High Resolution L3S Sea Surface Temperature product (SST_MED_PHY_L3S_MY_010_042) with a spatial resolution of 1/20° (Pisano et al., 2016; Embury et al., 2024). The assigned observation error standard deviation was 0.4° C.

Along track SLA data were sourced from the CMEMS SEALEVEL_EUR_PHY_L3_MY_008_061 product, which includes data from multiple altimeter missions at 7 km resolution. The assimilated quantity was the absolute dynamic topography (ADT), computed as a sum of the SLA and the mean dynamic topography (MDT). A bias correction was applied by subtracting the temporal mean of ADT and adding the MDT from the 2019-2022 ROMS free run, interpolated along the satellite tracks. To better constrain the free surface, each SLA observation was repeated one hour before and after its timestamp, following the assumption of slowly varying sea level signals as Zavala-Garay et al. (2014) and Levin et al. (2021).

Super-observations were created by averaging data within the same model grid cell (horizontally and vertically) and time window (hourly), as described by Moore et al. (2011c). A background quality control procedure was applied following Moore et al. (2013), rejecting observations when $d_i^2 > \beta^2(\sigma_{\text{b}}^2 + \sigma_{\text{o}}^2)$, where $d_i$ is the $i$-th component of the innovation, $\sigma_{\text{b}}^2$ is the background error at observation location, $\sigma_{\text{o}}^2$ is the observation error variance and $\beta$ is a coefficient that depends on the type of observation. Here $\beta$ was assigned a value 5 for all observation types meaning that observations with innovations that exceed five times the square root of the expected total error variance are rejected.

The spatial density of assimilated observations is reported in Figure 1b, while the distribution with depth and typology is reported in Figures 1c and 1d, respectively. The amount of observations per assimilation window throughout the year is reported in Figure 1e. In total, approximately $4.5 \cdot 10^6$ observations were assimilated during 2022, with a mean of $37.12 \cdot 10^3$

observations per assimilation cycle, a maximum of $49.15 \cdot 10^3$, a minimum of $16.29 \cdot 10^3$ and a standard deviation equal to $5.82 \cdot 10^3$. Further temporal and depth-dependent distributions by data source are presented in Figures 3 and 4.

## 2.3 Quantifying impact of observations

The properties of the adjoint model allow quantification of the impact that each assimilated observation has on a specific scalar metric $I(\mathbf{z})$, following the approach described by Langland and Baker (2004). Specifically, the increment in the metric $\Delta I(\mathbf{z}) = I(\mathbf{z}_a) - I(\mathbf{z}_b)$ due to DA (Moore et al., 2011b, 2017) can be approximated as:

$$\Delta I(\mathbf{z}) \approx \mathbf{d}^T \mathbf{K}^T \mathbf{M}_\mathbf{b}^T(t) \partial I / \partial \mathbf{z} \tag{3}$$

where $\mathbf{M}_\mathbf{b}(t)$ is the tangent linear model computed around the background solution $\mathbf{x}_b$. In this study, we consider the time-averaged volume transport integrated over a transect, expressed as: $I(\mathbf{z}) = \frac{1}{N_t} \sum_{n=1}^{N_t} \mathbf{h}_n^T \mathbf{v}_{\perp,n} = \overline{Tr}$ where $N_t$ is the total number of time steps, $\mathbf{v}_{\perp,n}$ is the velocity component orthogonal to the transect, and $\mathbf{h}_n$ is a vector comprising the contributions to cross-sectional area at each time step $n$. After replacing $\mathbf{K}$ by the approximation in equation 2, the impact $\Delta I(\mathbf{z})$ on transport becomes:

$$\Delta I(\mathbf{x}) \approx \frac{1}{N} \mathbf{d}^T \mathbf{R}^{-1} \mathbf{G} \mathbf{B} \mathbf{G}^T \mathbf{V}_m \mathbf{T}_m^{-1} \mathbf{V}_m^T \mathbf{G} \mathbf{B} \sum_{n=1}^{N_t} \mathbf{M}_\mathbf{b}^T(t_n) \mathbf{h}_n \tag{4}$$

The contribution of a single observation $y_i$ is readily computed from the dot-product of the $i$-th component of the innovation $\mathbf{d}$ and the $i$-th component of the vector $\mathbf{g} = \mathbf{K}^T \mathbf{M}_\mathbf{b}^T(t) \partial I / \partial \mathbf{z}$. Therefore, the averaged increment for an assimilation window is $\Delta I(\mathbf{z}) = \sum_{i=1}^{N_{obs}} d_i \cdot g_i = \Delta \overline{Tr}$. Moreover, the total observation impact can be decomposed according to the control vector to the 4D-Var increments as $\Delta I(\mathbf{z}) = \mathbf{d}^T \cdot (\mathbf{g_x} + \mathbf{g_f} + \mathbf{g_b}) = \sum_{i=1}^{N_{obs}} d_i \cdot (g_{x,i} + g_{f,i} + g_{b,i})$ where $\mathbf{g_x}$, $\mathbf{g_f}$ and $\mathbf{g_b}$ represent the contributions to $\mathbf{g}$ from the initial conditions (IC), atmospheric forcing (AF) and open boundary conditions (BC), respectively (Moore et al., 2011b; Levin et al., 2021).

In the present work, we focus on the volume transport across the Corsica Channel, evaluated along the transect between (9.4250E, 43.0204N) and (9.8369E, 43.0204N) reported in Figure 1a. We quantify the contribution of the different observation platforms to the resulting transport increments and their respective influence on each component of the 4D-Var control vector.

## 3 Results

### 3.1 Performance of the DA system

A first indication of the effectiveness of the assimilation procedure in fitting the model to observations is provided by the behavior of the cost function $J$ (Moore et al., 2011a). Figure 2a shows the initial and final values of $J$, along with the ratio of the final to initial observation-related cost component, $J_o = \frac{1}{2}[\mathbf{y} - H(\mathbf{x})]^T \mathbf{R}^{-1}[\mathbf{y} - H(\mathbf{x})]$. Final values of $J$ are consistently lower than the initial ones, with the ratio $J_{o,fin}/J_{o,ini}$ ranging between 0.2 and 0.6.

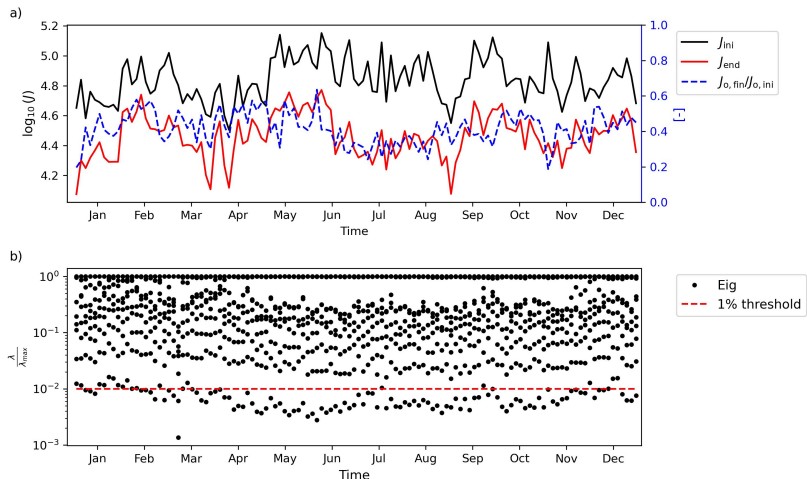

**Figure 2.** a) Time evolution of the initial cost function (black line), its value after data assimilation (red line), and the ratio of the final to the initial value of the part of the cost function related to observations (blue dashed line). b) Ratio of eigenvalues to the maximum eigenvalue associated with the matrix $\mathbf{T}_m$ (black dots) and 0.01 threshold (red dashed line), for each assimilation window.

Insight into potential overfitting to the observations is provided by analyzing the eigenvalues of the $\mathbf{R}$-preconditioned stabilized representer matrix $\tilde{\mathbf{P}}$ (McIntosh and Bennett, 1984; Moore et al., 2018, 2021). The majority of the increment $\delta\mathbf{z}$ is typically associated with the smallest eigenvalues, which may introduce spurious noise and degrade the solution. Therefore, a useful rule-of-thumb to prevent overfitting is to retain, for each assimilation cycle, only the contributions from eigenvalues $\lambda$ satisfying $\lambda/\lambda_{\max} > 0.01$ (McIntosh and Bennett, 1984). Since the eigenvalues of $\tilde{\mathbf{P}}$ match those of the tridiagonal matrix $\mathbf{T}_m$, they can be directly computed as a diagnostic during each 4D-Var cycle. Figure 2b shows that the number of inner loops used does not lead to significant overfitting, with at most only the final eigenvalue falling below the $0.01\lambda_{\max}$ threshold.

Model performance in reproducing observations is shown in Figure 3 and Figure 4 for the FR, FC and AN runs. Figure 3 shows the time evolution of the root mean square error (RMSE), computed over individual assimilation windows, for spatially distributed variables: surface velocities from HFR, $u_{\mathrm{HFR}}$ and $v_{\mathrm{HFR}}$, CORA in situ temperature and salinity $T_{\mathrm{CORA}}$ and $S_{\mathrm{CORA}}$, and satellite-derived SST and SLA. RMSE is calculated as $\frac{1}{N_{\mathrm{obs}}^{\mathrm{w}}} \sum_{i=1}^{N_{\mathrm{obs}}^{\mathrm{w}}} \sqrt{y_i - y_i^{\mathrm{m}}}$, where $N_{\mathrm{obs}}^{\mathrm{w}}$ is the number of observations for a specific observed variable during an assimilation window, $y_i$ is the $i$-th observation value and $y_i^{\mathrm{m}} = H(x)$ the model counterpart in observation space. The number of observations per window is also shown as a gray bar on the secondary y axis. The assimilation improves both FC and AN runs with respect to the FR across all variables (Figure 3g). For surface currents measured by the HFR, a difference between the three runs, with the FC falling between FR and AN is clearly observable (Figure 3a and b). The weighted average RMSE reduction compared to FR is between 10.1% ($u$) and 17.7% ($v$) for the forecast ($\overline{\Delta\mathrm{RMSE}^*_{\mathrm{FC}}}$) and between 33.7% ($u$) and 40.8% ($v$) for the analysis ($\overline{\Delta\mathrm{RMSE}^*_{\mathrm{AN}}}$) (Table 1). CORA in-situ observations of $T$ and $S$ (Figure 3c and d) exhibit periods where FC and AN are similar (especially for salinity Figure 3d) and with a quite large variability in the number of observations, which decreases in the second half of the year. The RMSE reduction for FC and AN

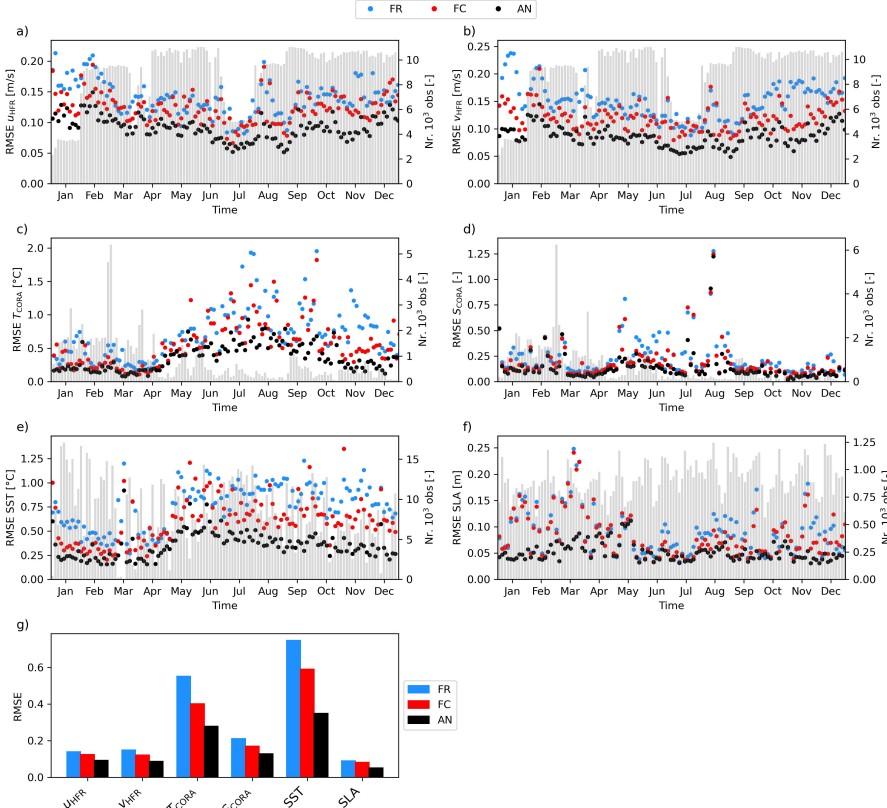

**Figure 3.** Time series of the RMSE between modeled and observed values for the free run (FR) in blue, forecast (FC) in red and analysis (AN) in black. Grey bars represent the amount of assimilated observations during each assimilation window .a) and b) u- and v-velocity components from HFR, respectively; c) and d) in situ temperature and salinity from the CORA dataset, respectively; e) SST, f) SLA. In g) the average RMSE weighted on the number of observation per assimilation window for the variables analyzed is reported.

in temperature, $\overline{\Delta\mathrm{RMSE}^*}_{\mathrm{FC}}$ and $\overline{\Delta\mathrm{RMSE}^*}_{\mathrm{AN}}$ for $T_{\mathrm{CORA}}$ are equal to 25.6% and 46.7%, respectively. A similar result holds for $S_{\mathrm{CORA}}$ (Table 1). SST and $T_{\mathrm{CORA}}$ show a notable RMSE increase for the FR, and less so for the FC, starting in mid-May, while $\mathrm{RMSE}_{\mathrm{AN}}$ tends to remain more stable (Figure 3c and e). Reduction in error ranges from 21.2% to 52.5% for SST and from 25.6% to 46.7% for $T_{\mathrm{CORA}}$, for FC and AN, respectively (Table 1). For the SLA the improvement due to the DA procedure is evident throughout the year for the AN, whereas for the FC it is more pronounced from the second half of June (Figure 3f). Indeed, the error reduction is 8.3 % for the forecast and 33.6% for the analysis (Table 1).

For observations collected by fixed platforms (those from the JERICO CoCM and the EMSO platforms), the vertical distribution of RMSE is shown in Figure 4 (panels a to d). Also in this case, the data assimilation procedure leads to a general improvement in the model skills for both the AN and FC fields across all observed variables (Figure 4e). For the current velocities observed by the CoCM, the RMSE reduction is more pronounced for the meridional ($v$) component than for the zonal ($u$) component, with the $v$ velocity reaching a maximum $\overline{\Delta\mathrm{RMSE}^*}_{\mathrm{AN}}$ reduction of 62.4% (Table 1 and Figure 4a and b). In

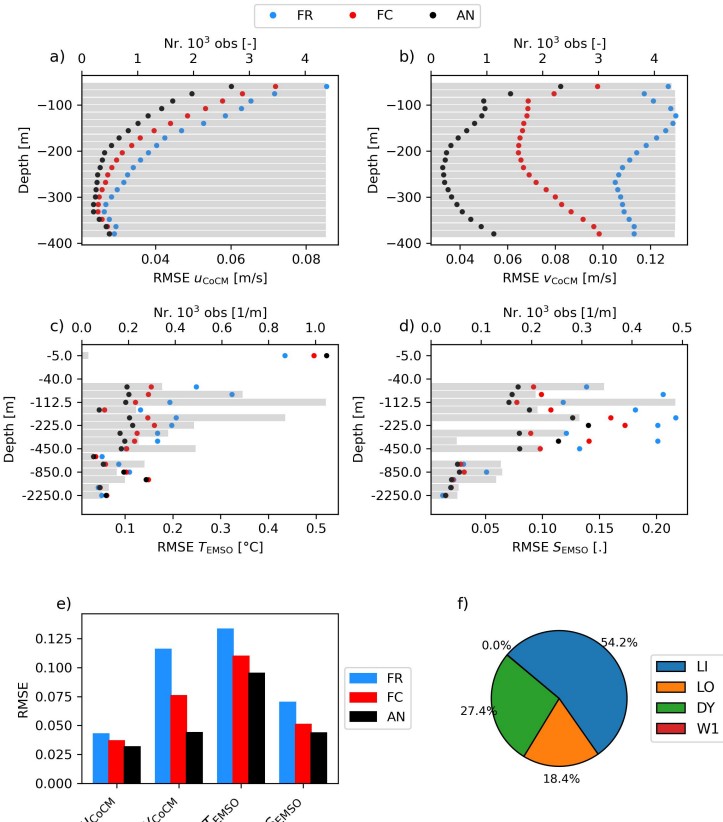

**Figure 4.** Distribution with depth of the RMSE between modeled and observed values for the free run (FR) in blue, forecast (FC) in red and analysis (AN) in black. Gray bars represent the number of assimilated observations at each depth; a) and b) $u$- and $v$-velocity components from the CoCM, respectively; c) and d) in situ temperature and salinity from the EMSO dataset, respectively. e) Average RMSE weighted on the number of observation per assimilation window for the variables analyzed is reported. f) Quantitative distribution of observations per platform (Figure 1).

situ temperature and salinity observations from the EMSO platforms show an uneven vertical distribution (Figure 4c and d), and a variable number of observations across platforms (Figure 4f). The RMSE reduction for both FC and AN is more evident above 450 m depth, while differences between the three simulations become less significant below this depth, down to 2250 m

(Figure 4c and d). The $\overline{\Delta\mathrm{RMSE}^*}$ is larger for the $S_{\mathrm{EMSO}}$ than for the $T_{\mathrm{EMSO}}$ (Table 1).

The improvement obtained through DA was also evaluated using the Pearson's correlation coefficient, $r = \frac{\sum_{i=1}^{N_{\mathrm{obs}}^{\mathrm{w}}}[y_i - \mathrm{avg}(\mathbf{y})][y_i^{\mathrm{m}} - \mathrm{avg}(\mathbf{y}^{\mathrm{m}})]}{\mathrm{std}(\mathbf{y}) \cdot \mathrm{std}(\mathbf{y}^{\mathrm{m}})}$, and the global mean difference (BIAS). The change in global BIAS ($\overline{\Delta\mathrm{BIAS}}$) and in mean correlation ($\overline{\Delta r}$) between the different model runs are reported in Table 1, further confirming the overall positive impact of data assimilation on model performance.

| Variable | $\overline{\Delta RMSE}^*$ [%] | | $\overline{\Delta r}$ [-] | | $\overline{\Delta BIAS}$ [m.u.] | | std(y) [m.u.] |
|---|---|---|---|---|---|---|---|
| | FC | AN | FC | AN | FC | AN | |
| $u_{HFR}$ | -10.1 | -33.7 | 0.14 | 0.38 | 0.005 | -0.002 | 0.127 |
| $v_{HFR}$ | -17.7 | -40.8 | 0.20 | 0.38 | 0.013 | -0.026 | 0.136 |
| $u_{CoCM}$ | -13.3 | -23.6 | 0.21 | 0.36 | 0.004 | 0.003 | 0.022 |
| $v_{CoCM}$ | -34.1 | -62.4 | 0.15 | 0.38 | -0.072 | -0.078 | 0.091 |
| $T_{CORA}$ | -25.6 | -46.7 | 0.02 | 0.04 | -0.094 | -0.109 | 4.017 |
| $S_{CORA}$ | -19.2 | -39.4 | 0.07 | 0.20 | -0.023 | -0.030 | 0.463 |
| $T_{EMSO}$ | -10.1 | -18.7 | 0.16 | 0.21 | -0.036 | -0.034 | 0.324 |
| $S_{EMSO}$ | -15.2 | -22.4 | 0.31 | 0.41 | -0.021 | -0.028 | 0.064 |
| SST | -21.2 | -52.5 | 0.06 | 0.17 | -0.071 | -0.094 | 5.220 |
| SLA | -8.3 | -35.6 | 0.04 | 0.08 | 0.015 | 0.001 | 0.053 |

**Table 1.** Columns 2 and 3 show the average relative difference of RMSE between the forecast and freerun (FC), and the analysis and freerun (AN), weighted on the number of observation for each assimilation window for each observation type in column 1; negative values indicate improvement and vice-versa. Columns 4 and 5 show the average increase/decrease in Pearson's $r$ correlation between forecast and freerun (FC), and analysis and freerun (AN), weighted on the number of observation for each assimilation window; positive values indicate improvement and vice-versa. Columns 6 and 7 show the difference in the mean total bias (in absolute value) between forecast and freerun (FC), and analysis and freerun (AN). For the $\overline{\Delta BIAS}$ each value have the measurement unit of the associated variable; negative values indicates improvement and vice-versa. Last column contains the standard deviation of observations.

The correlation between model output and observations consistently increases following the DA procedure, as indicated by positive values of $\overline{\Delta r}$ (Table 1). In contrast, the global BIAS shows a slight increase in some cases relative to the FR. However, BIAS values remain generally smaller than the standard deviation of the observed variables, regardless of wether model performance improves or slightly deteriorates. For instance, the increase in $\overline{\Delta BIAS}$ for $u_{CoCM}$ ranges between 0.003 and 0.004 m/s, compare to the standard deviation of 0.037 m/s in the observations. Similarly, for SLA, $\overline{\Delta BIAS}$ varies between 0.001 and 0.015 m, while the standard deviation of the observed SLA is 0.053 m.

Figure 5 compares the observed and modeled northward velocity component of the CoCM time series at three different depths to evaluate DA performance in reproducing transport through the Corsica Channel. Figure 5a-c shows better agreement in AN than FR. Winter velocity peaks are generally captured, albeit the FR shows surface overestimation and bottom under-estimation in late 2022. Annual mean profiles of northward velocity across the channel (Figure 5d, e) confirm these trends: FR overestimates velocity, while AN aligns well with observations. Negative velocity areas are reduced in AN. Differences between AN and FC are minor and are not shown.

### 3.2 Observation impact on Corsica Channel transport

Time series of total northward transport through the CC for the three runs (FR, FC and AN) are shown in Figure 6a. The annual averages are 0.49, 0.31 and 0.28 Sv, respectively.

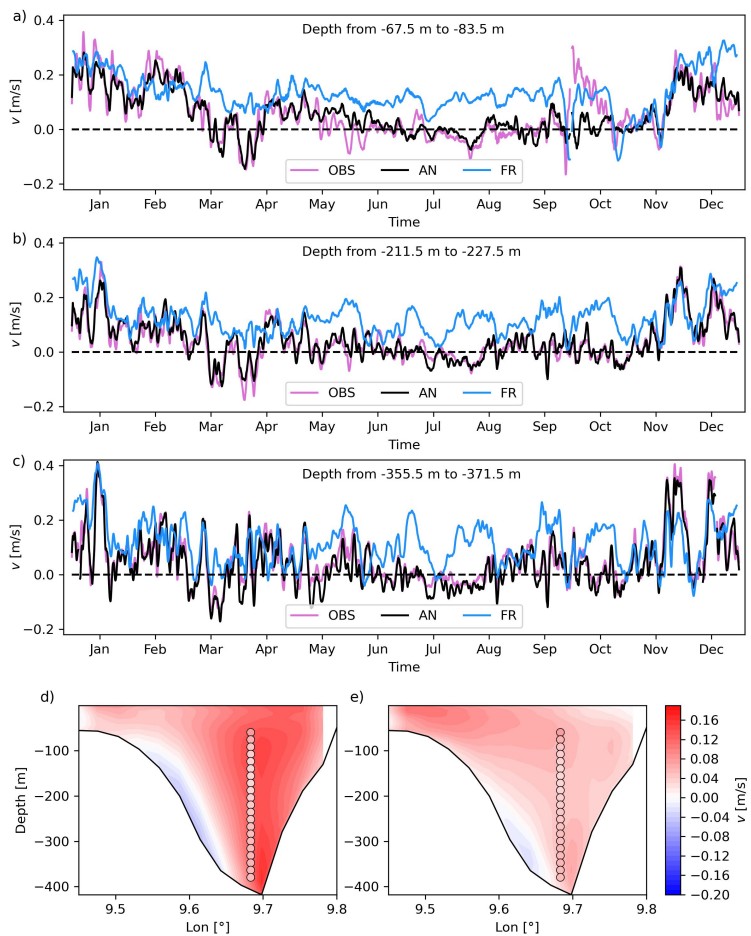

**Figure 5.** Time evolution of the observed (violet), and modeled by FR (light blue) and AN (black) runs of the $v$-component of the velocity at the CoCM at different depth intervals: a) from -67.5 m to -83.5 m; b) from -211.5 m to 227.5 m; c) from -355.5 m to -371.5 m). Yearly averaged value of the $v$-component of the velocity at the transect in correspondence of the CC mooring for FR d) and AN e), together with yearly averaged observed values (colored dots).

Both DA-informed simulations (FC and AN) show a net reduction in northward transport compared to the free run, although all three simulations capture expected seasonal variability, characterized by higher transport in winter and lower values in summer. The most notable differences occur in summer season and early fall: between June and October, average transport is 0.29 Sv in the FR and only 0.03 Sv in the AN. For the remaining part of the year, averages are 0.63 Sv and 0.45 Sv, respectively. Nonetheless, these results are not necessarily generalizable to other years, as DA could potentially increase transport in different
conditions.

Figure 6b compares the increment in time-averaged transport estimated using the tangent linear approximation ($\Delta \overline{Tr}_{\text{TL}}$, Equation 4) with that computed directly from the difference between the non-linear runs AN and FC ($\Delta \overline{Tr}_{\text{NL}}$). The good agreement between the two time series confirms that the tangent linear approximation holds well within the three-day assimilation window. Increments in average transport during an assimilation window range between -0.45 and 0.39 Sv, with a mean
of -0.03 Sv and a standard deviation of 0.16 Sv.

The contribution of each observation type to changes in the total northward transport is shown in Figure 6c, as detailed in Section 2.3. Observations from the CoCM have the largest overall impact on the transport increment, followed by HFR, SST, in situ $T$ and $S$, and SLA. The relative impact of observation types also varies seasonally: CoCM data dominate during winter and autumn, while HFR observations have a greater influence during spring and summer. No clear seasonal pattern is observed
for the remaining data sources.

Figure 6d quantifies the contribution of different components of the 4D-Var control vector (initial condition, boundary conditions, surface fluxes of heat, freshwater and momentum) to the CC transport increment $\Delta \overline{Tr}$. The southern boundary corrections dominate throughout the year, indicating that transport is influenced by large-scale basin dynamics beyond the model's current spatial extent. Initial conditions have a limited role, except at the beginning and end of the year, while surface
forcing becomes more influential from June onward.

An indicator for assessing the effectiveness of the DA procedure is a scatter plot of the contribution of each observation $d_i \cdot g_i$ (i.e., its impact on $\Delta I$) versus the corresponding innovation element $d_i$, as illustrated in Figure 7. These scatter plots can also be interpreted as contingency diagrams, with the percentage of points in each quadrant indicated.

Overall, the points in Figure 7 are approximately evenly distributed between positive and negative innovations for all ob-
servation types, except for in situ salinity (Figure 7f). This can be viewed as an indication that a clear and evident bias is not present in these observed components of the state vector, but for the in situ salinity, for which a non symmetric distribution for positive and negative innovation values is present (44.4% negative; 55.7% positive). The "butterfly" shape of each scatter plot indicates that observations with small innovations tend to produce only minor changes in CC transport. This behaviour is expected, as observations that require only small corrections on the background state should not significantly alter transport
estimate. Conversely, observations associated with larger innovations, exhibit a wider range of impacts on CC transport.

By and large, for HFR, SST, SLA, CoCM and in situ salinity, impacts are approximately symmetrically distributed around $d_i \cdot g_i = 0$ , meaning that these data sources contribute both positively and negatively to CC transport adjustments. In contrast, in situ temperature observations show a skewed distribution, with approximately 61% of the points leading to a reduction in transport ($d_i \cdot g_i < 0$). Figure 7 also reveals a pronounced asymmetry in the distribution of points relative to $d_i = 0$ for SLA and

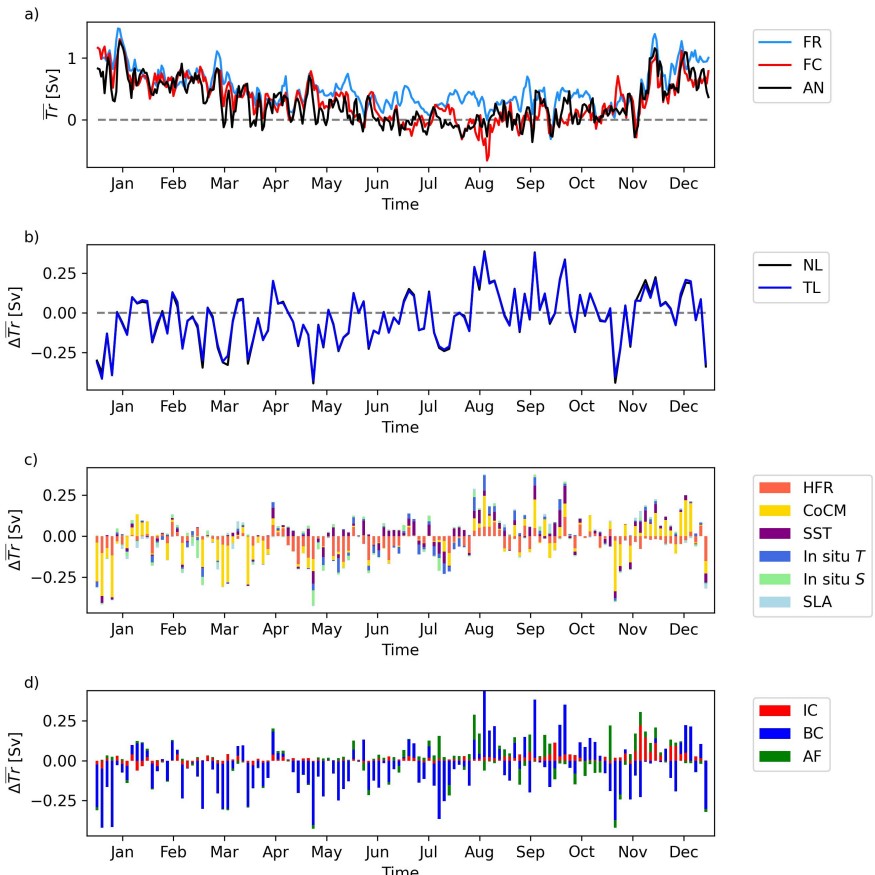

**Figure 6.** a) Total transport through the Corsica Channel for the freerun (light blue line), forecast (red line) and analysis (black line). b) Increment in total transport calculated by the difference between the analysis and background runs (black line) and through the tangent linear approximation (blue line). c) Impact of the different observation typologies in increasing/decreasing the total transport: HFR (orange bars), in situ CC mooring (yellow bars), SST (purple bars), in situ $T$ (blue bars), in situ $S$ (light green bars), SLA (light blue bars). d) Impact on increase/decrease of the total transport split by the correction applied to the background run: initial condition (IC, red bars), boundary conditions (BC, blue bars), atmospheric forcing (AF, green bars).

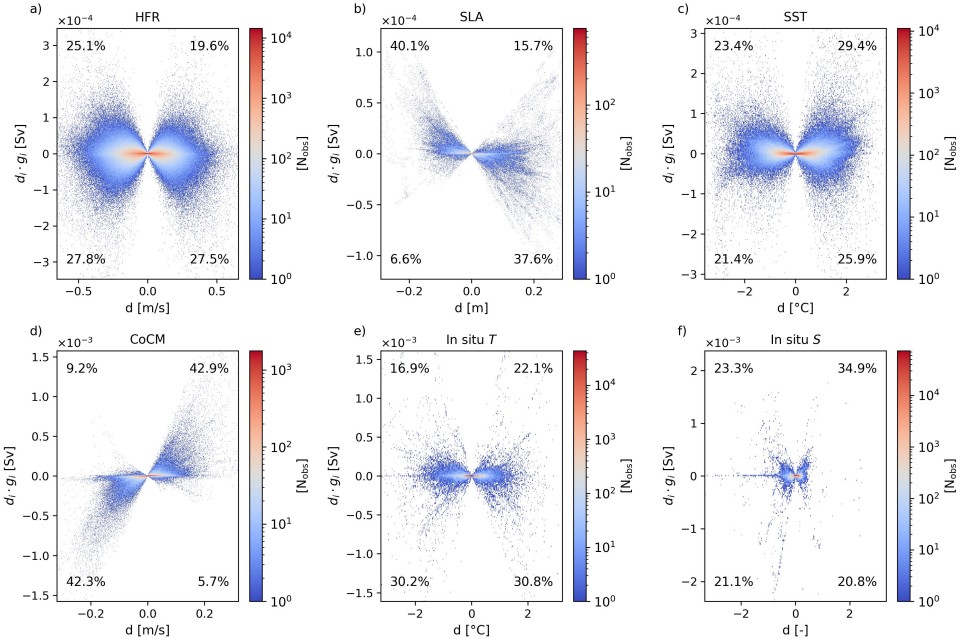

**Figure 7.** Density scatter plot of innovation **d** versus single increments $d_i \cdot g_i$ for different typology of observations: a) high-frequency radars, b) sea surface height, c) sea surface temperature, d) Corsica Channel mooring, e) in situ temperature, f) in situ salinity.

CoCM observations. Specifically, negative SLA innovations ($d_i < 0$) typically lead to an increase in CC transport ($d_i \cdot g_i > 0$), and vice versa. The opposite pattern is observed for CoCM data. While such a behavior is clear for the CoCM $v$-velocity component, further analysis is needed to fully understand the underlying mechanism driving the SLA signal in the DA system.

An additional metric for evaluating observation impacts is the root mean square (RMS) of transport increments, calculated for both the aggregate effect of each observation type and for the average contribution of a single observation (Levin et al., 2021). The Global RMS for each observation typology is calculated as $\sqrt{\frac{1}{N_w} \sum_{i=1}^{N_w} \Delta \overline{Tr}_i^2}$, where $N_w$ is the total number of assimilation windows and $\Delta \overline{Tr}_i$ is the average transport increment associated with the $i$-th assimilation window. The average RMS impact per individual observation type (Datum) is given by $\sqrt{\frac{1}{N_{obs}} \sum_{i=1}^{N_{obs}} (d_i \cdot g_i)^2}$, where $N_{obs}$ is the total number of observations of a given type. Results for both metrics are reported in Table 2.

In situ velocity observations from the CoCM are the most influential in constraining the CC transport, especially when considering the average impact per observation. Aggregated over time, HFR data emerge as the second most effective data source. However, each HFR observation is approximately 3-5 times less impactful than in situ $T$ and $S$ measurements, and about 7 times less impactful than CoCM data. This can be explained by the fact that in situ data directly affect the whole water column, whereas surface data can only influence it indirectly, through the adjoint. While SST observations have greater overall impact than SLA on aggregate, the impact of individual observations of each of the two types is similar (Table 2) in agreement with the different amount of observations $N_{obs}$.

| Variable | RMS Impact [Sv] | | |
|---|---|---|---|
| | Global $\cdot 10^{-2}$ | Datum $\cdot 10^{-5}$ | $N_{\text{obs}} \cdot 10^3$ |
| HFR | 6.19 | 2.02 | 2194.8 |
| CoCM | 9.03 | 14.80 | 183.7 |
| In situ $T$ | 2.64 | 5.45 | 625.2 |
| In situ $S$ | 1.99 | 9.58 | 376.2 |
| SST | 3.63 | 1.68 | 1037.9 |
| SLA | 1.07 | 1.52 | 111.2 |

**Table 2.** Root mean square impact for the different observation typologies, both globally (Global refers to the effect on transport of all observations belonging to a certain typology), and considering the average contribution of a single observation (Datum refers to the average effect on transport of a single observation belonging to a certain typology). The last column reports the total number of observations per type.

## 4 Discussion

### 4.1 Changes in CC transport and overall circulation

Recalling that the yearly averaged transport values that we obtained at the CC for FR, FC and AN are equal to 0.49, 0.31 and 0.28 Sv, respectively, these values are lower than previous estimates based on observations, ranging from 0.54 Sv to 0.71 Sv (Astraldi and Gasparini, 1992; Astraldi et al., 1994), and numerical models, such as 0.49 Sv in Sciascia et al. (2019) and 0.5 Sv in Béranger et al. (2005), the latter based on a multi-decadal time series. However, reanalysis by de La Vara et al. (2019) and Barral et al. (2024) obtained an average transport around 0.33-0.34 Sv and 0.35 ±0.365 Sv, respectively, considering the whole transect including the CC and the area from the Island of Capraia to the Italian coast.

Part of the discrepancy with observational estimates may stem from the assumption of negligible longitudinal variability in the northward velocity across the channel that is not fully supported by our model results (Figure 5e and Figure 8d, e) and was also observed by previous numerical studies (Sciascia et al., 2019). Interestingly, the two reanalyses of de La Vara et al. (2019) and Barral et al. (2024) obtained an annual average transport comparable to our result by considering the whole Tyrrhenian transect. These values lower than those from previous literature can be ascribed to a possible negative annual averaged transport for the area between the Island of Capraia and the Italian Coast, or to a more appropriate representation of the flow in the area.

The transport across the whole transect linking the Tyrrhenian and Ligurian Seas, and the interaction between the two sub-transects, also in light of basin-scale mass budgets, deserve further studies. Furthermore, additional analysis, particularly over a longer time period, will help determine whether the year 2022 reflects inter-annual variability (Vignudelli et al., 1999), or is the indication of a longer-term trend.

Comparing the non-assimilative (FR) and the assimilative (FC and AN) runs it is important to stress that the latter are the cumulative result of corrections applied to the background by DA over each assimilation window. Observing Figure 5, it is clear that DA systematically reduces the northward velocity in the channel to better match the observations, leading to a corresponding reduction in total transport (Figure 6a). However, Figure 6b, shows that northward transport reduction is dominant up to mid-August 2022, after which positive transport increments become more frequent than negative ones.

This evolving pattern is further dissected in Figure 6c: from January to early April, CoCM data primarily drive the transport corrections; between April and early August, HFR data become more influential, despite not being directly located at the CC transect, where the transport is computed. Figures 8a and c show the average velocity increment $\delta v$ during the periods dominated by the JERICO facilities, CoCM (from 01 January to 09 April), and HFR data (from 22 April to 11 August), respectively. CoCM observations tightly constrain the flow near their location but with the side effect of accelerating the northward flow in the surface western part of the transect. In fact, the $\delta v$ values, averaged over the whole year, are characterized by an overall reduction for the most part of the CC transect and a slight increase at the upper western flank (not shown). This may be the result of insufficient corrections to the boundary conditions, redirecting the volume transport to unconstrained portions of the domain (Figure 8a, Figure 5e). Interestingly, it can help to explain why the total transport (Figure 6a) is less affected than the velocity alone (Figure 5a, b, c) when the FR and the AN runs are compared. Figures 5d and e show that while the northward flow weakens across most of the transect in AN with respect to FR, particularly on the eastern side (that observed by the mooring), a partial compensation occurs through a weaker (in absolute value) southward flow on the western flank and the already mentioned slightly northward acceleration at the upper western flank.

Figure 8b, d, f show the average spatial increment in transport per unit width, defined as $\Delta F(x,y)_{\text{avg}} = \sqrt{\Delta F_{u,\text{avg}}^2 + \Delta F_{v,\text{avg}}^2}$, where $\Delta F_{u,\text{avg}} = \frac{1}{T_f - T_i} \int_{T_i}^{T_f} \int_h \delta u \, dz \, dt$ and $\Delta F_{v,\text{avg}} = \frac{1}{T_f - T_i} \int_{T_i}^{T_f} \int_h \delta v \, dz \, dt$. Here, $h$ is water depth, and $T_i$, $T_f$ define the time interval, and the total transport can be recovered multiplying $\Delta F_{u,\text{avg}}$ and $\Delta F_{v,\text{avg}}$ by the cell widths $\Delta y$ and $\Delta x$, respectively (we prefer to consider this quantity rather than depth averaged velocity since the latter is independent of the volume of water moved). CoCM related transport increments show a pronounced southward pattern (Figure 8a and b), while HFR-driven increments manifest as surface- and bottom-intensified reductions in velocity, with weaker overall transport changes (Figure 8c and d). Furthermore, $\Delta F(x,y)_{\text{avg}}$ in Figure 8b is in general stronger than that of Figure 8d, despite the two show a similar pattern.

Another key period is from 18 November to 23 December, when CoCM-driven positive increments are counterbalanced by HFR-driven reductions (Figure 6c). The corresponding average increment in velocity at the Corsica Channel cross section $\delta v$ and the spatial distribution of $\Delta F(x,y)_{\text{avg}}$ for the eastern part of the modeled domain are reported in Figure 8e and f, respectively. Figure 8e reveals surface deceleration from HFR and subsurface acceleration from CoCM, with an additional northward acceleration provided by SST (Figure 6c), which is reasonably concentrated mainly at the surface. The corresponding average transport increment shows an east-west contrast, with acceleration near Corsica and deceleration across the Tuscany Archipelago (Figure 8f). This acceleration at the eastern flank of Corsica, and deceleration at the Tuscany coast, are present at the southern boundary, where an anticyclonic structure in the average transport increment is formed. We argue it is a modulation of the Bonifacio Gyre whose cyclonic structure, if weakened, tends to favor the transport in the CC for this particular current pattern. This happens in correspondence of a period (18 November - 23 December) where both IC and AF (other than BC) play a role in constraining the transport in the CC (Figure 6d) and it is indeed the atmospheric forcing that acts to modulate the intensity of the Bonifacio Gyre (Astraldi and Gasparini, 1994; Iacono and Napolitano, 2020).

This behavior exemplifies a key feature of 4D-Var: the adjoint model $\mathbf{M}_b^T$ uses the derivative $\partial I / \partial \mathbf{z}$ of the CC transport to inform the system what modifications are required at the control vector to influence $I$ during the 3 day assimilation window. For a more detailed explanation of the observation impact mechanism see Levin et al. (2021). Additional analyses might expand the

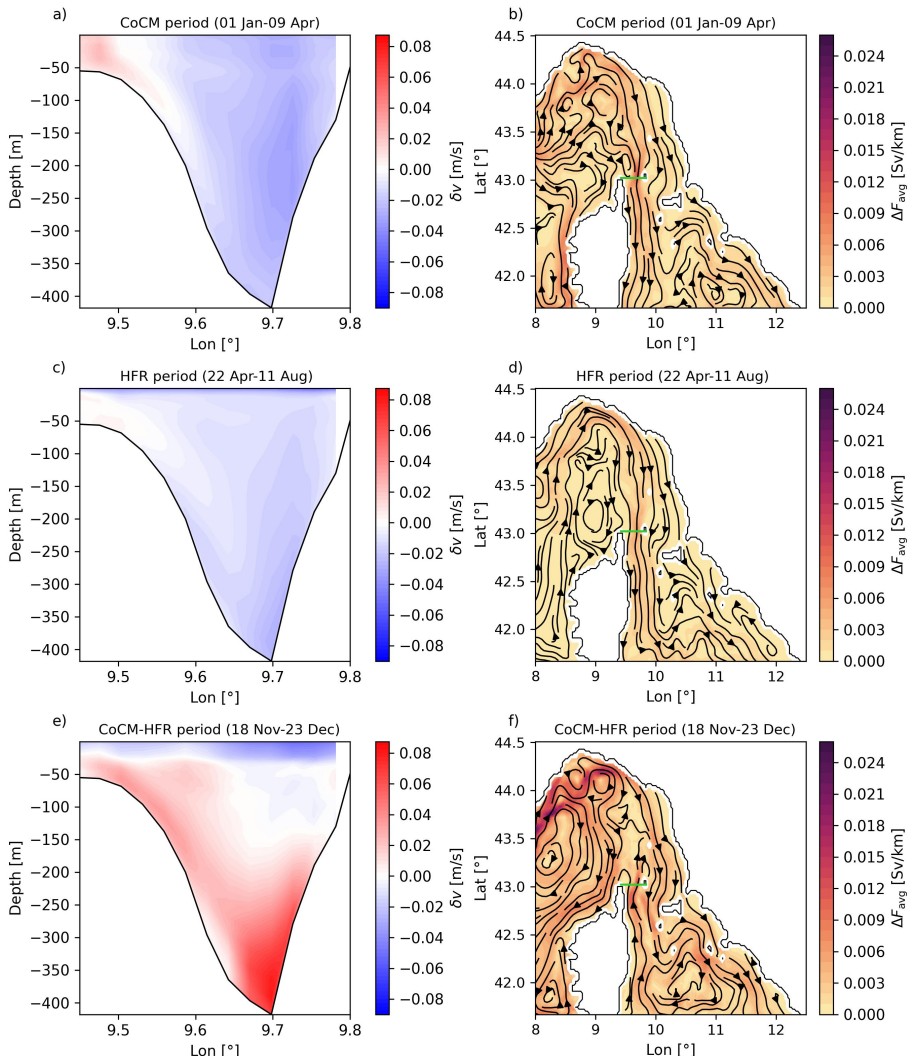

**Figure 8.** Time averaged values of the northward velocity increment $\delta v$ at the Corsica Channel for specific time intervals chosen with regards to Figure 6c: a) from 01 January to 09 April; c) from 22 April to 11 August; e) from 18 November to 23 December. Time averaged values of the increment in transport magnitude per unit width $\Delta F_{\text{avg}}$ for specific time intervals: b) from 01 January to 09 April; d) from 22 April to 11 August; f) from 18 November to 23 December. Streamlines are created on the basis of the vector field ($\Delta F_{u,\text{avg}}$, $\Delta F_{v,\text{avg}}$). The green line in panels b, d, f indicates the transect.

computational domain to include an additional portion of the domain which potentially influences the transport through the CC.

However, it is not straightforward to identify the extension which maximizes the impact of initial condition and atmospheric forcing at the expense of boundary condition. Furthermore, extending the domain to contain the whole Western Mediterranean basin would excessively increase computational costs.

If we focus on the physical processes affecting the performance of the DA system we can try to explain, for example, why the error reduction is more pronounced for the $v$ than the $u$ velocity component observed by the CoCM (Figure 4a, b, e and

400 Table 1). Since the latter has a lower observed average magnitude (0.008 m/s) and standard deviation (0.022 m/s) with respect to the former (0.05 m/s and 0.09 m/s, respectively), the DA has more chances to correct $v$ than $u$, also because we made the assumption that the background error covariance is proportional to the standard deviation of the state variables. Furthermore, being the flow orographically constrained, the eastward velocity ($u$) cannot reach large values and possible small variations are more hardly caught by the model.

Looking at Figure 3c and e, a regime shift is observable in the RMSE for $T_{\text{CORA}}$ and SST, which is larger for both FR and FC with respect to AN, starting approximately from mid-May. We argue it can be attributed to the marine heat wave (MHW) that affected the Western Mediterranean sea from mid-May to the end of summer (McAdam et al., 2024; Estournel et al., 2025), despite the regime shift in RMSE is present until the end of the year for both $T_{\text{CORA}}$ and SST. The effect of the MHW, although to a lesser extent, can be traced on the two RMSE peaks for $S_{\text{CORA}}$ in May and August, probably due to strong evaporation

processes (Figure 3d).

The difference in RMSE between AN and FR for $T_{\text{EMSO}}$ and $S_{\text{EMSO}}$, is remarkable for surface and intermediate water, while it reduces or disappears for deeper water (Figure 4c and d). It is likely explainable considering that AW, mAW and EIW are more sensitive than WMDW or TDW to seasonal changes that can be captured by DA.

Regarding the circulation, Figure 9 presents the time- and depth-averaged currents from FR and AN over two vertical layers:

surface to 200 m (AW or mAV), and 250 m to 450 m (EIW) (Artale and Gasparini, 1990).

DA slightly intensifies the NC, primarily by strengthening the WCC, which is barely noticeable in the FR, while the ECC flow is reduced (9a and b). This pattern can be seen as a shift from a Western Mediterranean gyre, which principally involves the Tyrrhenian and Provencal basins, toward a North-Western Mediterranean gyre, which partially substitutes the Tyrrhenian inflow with waters coming directly from the Algerian basin. The ECC behavior aligns with previous studies showing its high

variability (Astraldi et al., 1990; Astraldi and Gasparini, 1992), whereas the pronounced WCC signal was also found by Barral et al. (2024) and Ciuffardi et al. (2016). In the HFR domain, DA deflects the flow more westward, reducing the curvature observed in FR. The weak cyclonic structures between 4° and 8°E in FR are more pronounced in AN. Similar differences are seen in the EIW layer (Figure 9c and d).

We also looked for a peculiar pattern of the summer circulation in the area, that is the presence of the Ligurian Anticyclone,

for its potential role in modulating the ECC (Ciuffardi et al., 2016; Iacono and Napolitano, 2020). The determination of monthly averaged surface velocity fields allowed us to exclude the presence of the LA in the summer season, for both FR and AN. However, in October, a cyclonic feature north of Corsica was present in the AN and not in the FR run (not shown).

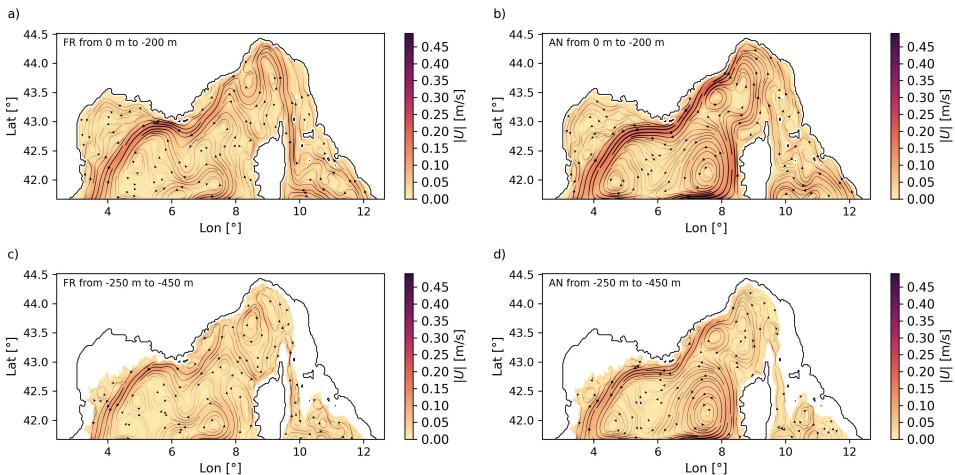

**Figure 9.** Depth and yearly averaged currents for different depth ranges as streamlines superimposed to the velocity magnitude $|U|$. a) freerun for the depth interval from the surface to -200; b) analysis for the depth interval from the surface to -200 m; c) freerun for the depth interval from -250 m to -450 m; d) analysis for the depth interval from -250 m to -450 m.

Significant current reversal episodes were not observed in the corresponding period, but the presence of the LA might have concurred to maintain the transport lower for the analysis than for the freerun (Figure 6a).

## 4.2 Insights into DA mechanisms

Observation impact can allow us to investigate in which ways DA modifies the model state with respect to a particular metric (transport across a section in our case) by optimally balancing between observations and background solution.

Table 2 reports the RMS transport increments considering each typology of observation (Global) and the single observation per typology (Datum). Apart from CoCM and SLA data, it seems that the impact of a single observation is inversely proportional to $N_{obs}$. It could be explained by considering that the larger the amount of observations, the higher the probability that some of them are redundant and do not provide additional information content. On the contrary, CoCM/SLA observations are characterized by large/small Global and Datum RMS, given a comparable $N_{obs}$ (Table 2), showing that the former are crucial to constrain ECC, whereas the latter have a small effect for the present modeling setup.

Developing more the analysis, we can even estimate the contribution of each observation to the transport increment through $\Delta \overline{Tr}_{\mathrm{TL}} = \mathbf{d}^T \cdot \mathbf{g}$. The presence of a seasonality in $\Delta \overline{Tr}$ when we consider observation typology (Figure 6c), that is roughly cold months controlled by CoCM while warm months until late autumn by HFR, could be explained by the corresponding seasonality in the CC transport. Small northward velocity values in warm months tend to produce small innovations $\mathbf{d}$ and, consequently, possible small $\mathbf{d}^T \cdot \mathbf{g}$. To better control the dynamic at the channel, DA relies on another source of observations that is HFR. In cold months, the opposite occurs and the role of CoCM observations is predominant.

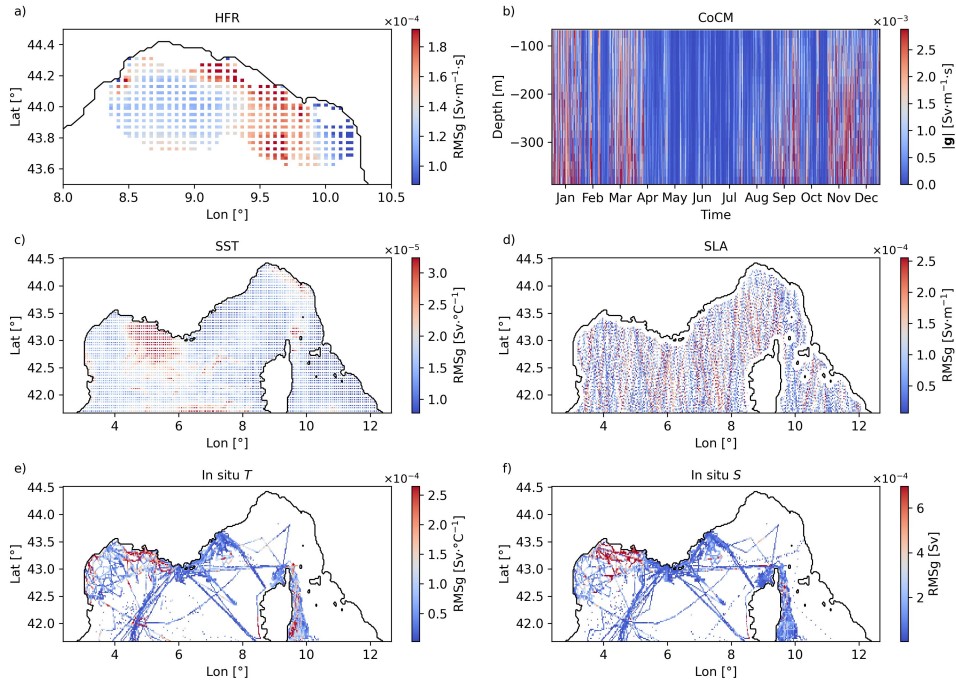

**Figure 10.** Spatial distribution of the root mean squared values of $\mathbf{g}$ (RMSg), for different observations: HFR a), SST c), SLA d), in situ $T$ e), in situ $S$ f). b) Absolute value of $\mathbf{g}$ as a function of observation depth and time for the $v$-velocity measured by the CoCM.

Moreover, the variability of the elements of $\mathbf{g}$ reveals the location of observations which have the greatest influence on the model. Figure 10 shows the spatial distribution of the root mean squared values of the elements of $\mathbf{g}$ (RMSg) calculated over each computational cell for each observation type.

RMSg for HFR peaks near the eastern part of the coverage region and close to the coast (Figure 10a), aligning with the CC's longitudinal range ($9.45°\text{E} - 9.8°\text{E}$).

The degree of sensitivity of the transport to SST and in situ T data is particularly high in the Gulf of Lion (GoL). Traces of the upwelling system induced by north-westerly winds (Odic et al., 2022) are detectable for both SST and in situ T (Figure 10c and e), whereas those associated to south-easterly winds at the western coast of the GoL (Odic et al., 2022) are mainly present for in situ T RMSg (Figure 10e). Similarly, it appears that salinity variations in the region of freshwater influence (ROFI) of the Rhône river can contribute to the evolution of the ECC (Figure 10f). High sensitivities in the area may be partly explained by the background-error covariance matrix $\mathbf{B}$ that we assumed to be proportional to the standard deviation of the freerun. For temperature, it displays large values at the surface in the eastern part of the GoL area and lower, but significant values, at the western part, not only at the surface (not shown). For salinity, large standard deviation values are restricted to the ROFI of the Rhône river (Fraysse et al., 2014) (not shown). However, the physical mechanisms through which upwelling/downwelling episodes or salinity variations in the GoL can influence CC transport, deserve and require further investigation.

Secondary hot-spots of RMSg for in situ $T$ and $S$ near the LI station and around the eastern coast of Corsica, especially for $T$, emphasize that distant observations can significantly affect CC transport (Figures 10e and f).

The large RMSg values associated with SST observations (Figure 10c) at the eastern coast of the Ligurian sea roughly correspond to the pattern observed for the RMSg associated with HFR data (Figure 10a). We believe the contribution of those SST observations to $\Delta \overline{Tr}$ is linked to that of HFR as their interaction is already observed in a similar 4D-Var application as reported by Bendoni et al. (2023).

The impact of SLA observations shows no clear spatial pattern (Figure 10d), although certain satellite tracks exhibit a larger RMSg values, suggesting that during specific periods the barotropic control on CC transport is significant. Indeed, if we plot the $g_i$ values as a function of time (not shown), we observe larger values at the year's start and end, suggesting that when the $\overline{Tr}$ is higher, corrections of the barotropic modes are larger.

Figure 10b shows a Hovmöeller diagram of the absolute values of the elements of $\mathbf{g}$ associated with the $v$-velocity component of observations from the CoCM. The sensitivity of the CC transport to these $v$-velocity observations appears relatively uniform with depth and displays a clear seasonal signal, consistent with the temporal variability seen in Figure 6d, where the contribution of CoCM data diminishes during late spring and summer. Although the vertical distribution of sensitivity is generally homogeneous, it tends to be more pronounced at the bottom. This may result from the fact that the $\mathbf{g}$ values are weighted by the $\mathbf{P}$ matrix (through the Kalman gain, see Equation 3), thus depending on $\mathbf{R}$, $\mathbf{B}$, and the spatial distribution of the other observations. As a result, the sensitivity of the $\overline{Tr}$ to CoCM data near the surface may be partially influenced by HFR, SST, SLA observations whereas at depth this effect is not present.

Focusing on the $g_i$ values helps to identify observation locations that are potentially influential for $\Delta \overline{Tr}$ and, by extension, for understanding the physical mechanisms driving transport in the CC. In contrast, analyzing the product $d_i \cdot g_i$ (i.e., individual increments) may mask this insight, as a large sensitivity $g_i$ could be masked by a small innovation $d_i$, or vice versa. A more targeted approach to quantify the influence of different regions or variables on the CC transport would be an adjoint sensitivity study (Moore et al., 2009; Veneziani et al., 2009). However, such an approach does not offer information about how observational data significantly shape the model solution.

For most assimilation windows the primary way DA constrains the CC transport is the modification of the boundary conditions (Figure 6d). By selecting those with the largest positive and negative transport increments largely attributable to BC modifications (90-th and 10-th percentiles of the $\Delta \overline{Tr}_{BC}$ distribution, respectively), we calculated the northward velocity increment at the southern boundary $\delta v$ (Figure 11a and b), the free surface height increment $\delta \eta$ (Figure 11c and d), and net heat flux increment $\delta Q_{\text{net}}$ (Figure 11e and f), between analysis and background, averaged over the two sets of assimilation windows. A positive $\Delta \overline{Tr}_{\text{BC}}$ is indeed associated with a northward-inducing $\delta \eta$ (Figure 11c) that is consistent with a clear northward barotropic acceleration on the Tyrrhenian side and a less intense southward barotropic acceleration, distributed over a much wider cross sectional area, at the Ligurian-Provençal basin side (Figure 11a). For the negative $\Delta \overline{Tr}_{\text{BC}}$ such a mechanism is more marked for both $\delta v$ and $\delta \eta$. The barotropic nature of these increments is confirmed by the almost uniform increment velocity distribution along depth (Figure 11a and b). Interestingly, the $\delta Q_{\text{net}}$ is characterized by a pronounced surface heating/cooling in the Ligurian-Provençal basin when the CC transport increases/decreases (Figure 11e and f). Since the ROMS model

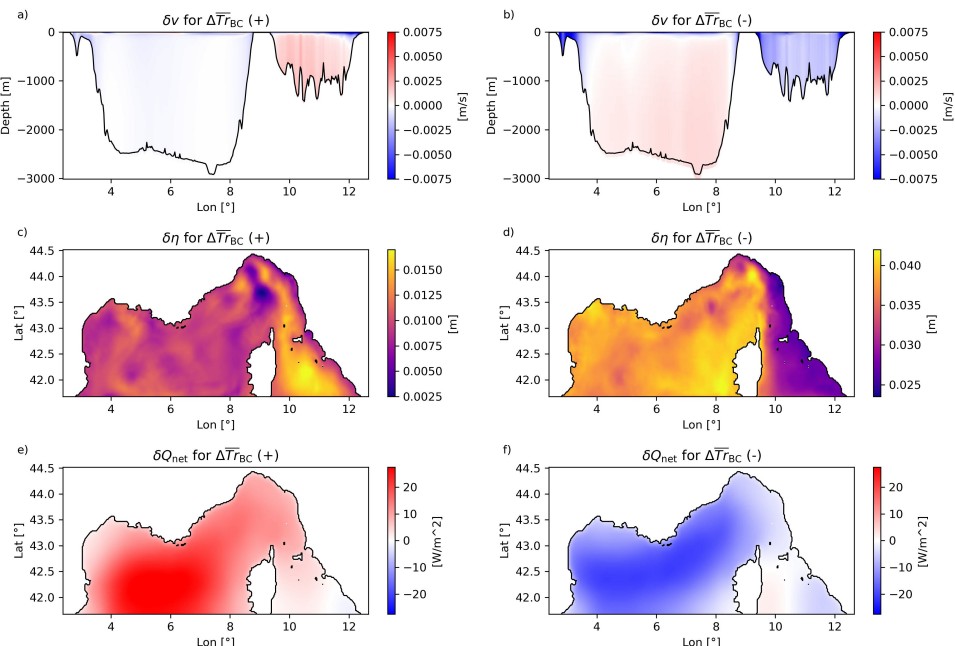

**Figure 11.** Time averaged difference between analysis and background over specific assimilation windows for the fields northward velocity $v$ at the boundaries, free surface $\eta$ and net heat flux $Q_{\text{net}}$ in case the increment in transport $\Delta \overline{Tr}_{\text{BC}}$ is positive, a), c), e) and in case it is negative b), d), f).

adopts the Boussinesq approximation (Cushman-Roisin and Beckers, 2011), such a mechanism cannot be related to steric adjustments, but rather to a modification to the baroclinic pressure field to maintain the consistency between the barotropic correction at the boundaries and observations in the interior. Additional analysis is required to make explanatory hypotheses at the base of the observed mechanism.

Astraldi and Gasparini (1992) proposed that CC dynamics are governed by atmospheric conditions over the western part of the Ligurian-Provençal basin, where winter heat loss enhances the steric gradient between the Tyrrhenian and Ligurian-Provençal basins, driving Tyrrhenian waters into the Ligurian Sea and strengthening the ECC. Our results cannot directly mimic the processes described by Astraldi and Gasparini (1992), however, focusing now on the largest positive and negative transport increments attributable solely to the correction to atmospheric forcing, $\Delta \overline{Tr}_{\text{AF}}$ (95-th and 5-th percentiles of the $\Delta \overline{Tr}_{AF}$ distribution, respectively), we observe that $\delta \eta$ (Figure 12a and b) and $\delta Q_{\text{net}}$ (Figure 12c and d) align with the mechanism proposed by Astraldi and Gasparini (1992): an increase (decrease) in transport is triggered by a differential surface cooling (warming) with associated eastward (westward) positive gradient in the free surface increment. Even in this case, additional analyses are required to delve into the mechanism responsible for the agreement with literature results.

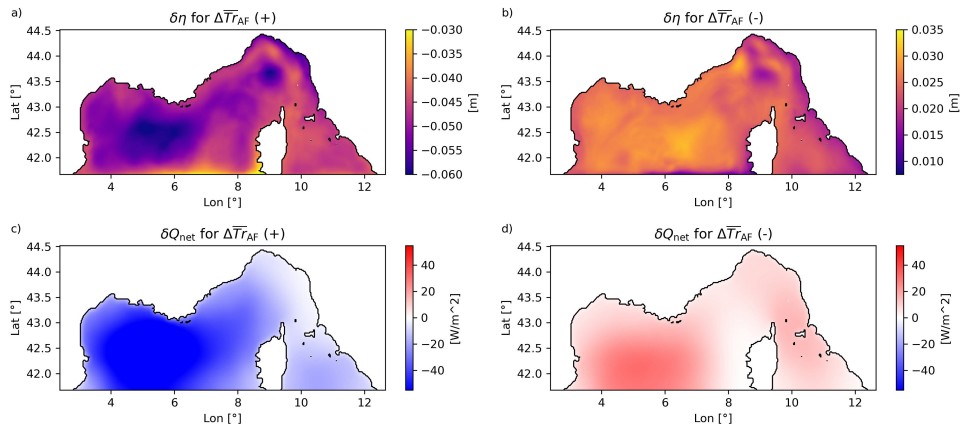

**Figure 12.** Time averaged difference between analysis and background over specific assimilation windows for the fields free surface $\eta$ and net heat flux $Q_{\text{net}}$ in case the increment in transport $\Delta \overline{Tr}_{\text{AF}}$ is positive, a), c) and in case it is negative b), d).

## 5 Conclusions

The ECC is a key feature of the north-western Mediterranean Sea circulation, transporting Tyrrhenian waters into the Ligurian
Sea. In this study, we used the ROMS 4D-Var system to assess how various observations constrain the transport through the
CC and to investigate the mechanisms underlying its variability. Surface velocities from HFR, SST, SLA, in situ temperature,
salinity and velocity profiles-measured directly in the CC were assimilated for the year 2022 using 3 days assimilation windows,
with one outer loop and nine inner loops.

The results from the analysis (AN) and forecast (FC) runs show significant improvements in the representation of circulation,
compared to the free run (FR) without data assimilation. RMSE reductions ranged from 8.3% to 34.1% in the FC and from
18.7% to 62.4% in the AN. Pearson correlation coefficients ($r$) increased by 0.04 to 0.31 in the FC and by 0.04 to 0.41 in the
AN. While BIAS values in the forecast exhibited both increases and decreases, reductions were dominant in the analysis.

After assimilation, the annual mean transport through the CC decreased from 0.49 Sv in the FR to 0.31 Sv in the FC and 0.28
Sv in the AN. The comparison between the AN and FR indicates that the ECC deceleration is accompanied by an acceleration
of the WCC, occurring both at the surface (AW) and at intermediate depths (EIW).

The reduction in CC transport is mainly attributed to the assimilation of CoCM data, which generally slow down the ECC
during the first part of the year, and then contribute to its acceleration from September onward. HFR data represent the second
most influential source for constraining CC transport, particularly contributing to the reduction during late spring and summer.
SST, in situ temperature and salinity, and SLA follow in importance. The various data types influence the northward velocity
pattern across the CC differently, sometimes in competition, as in the case where HFR data tend to decelerate surface flow,
while CoCM data act to accelerate flow at depth. Transport sensitivity analyses reveal that circulation features located far from
the CC, such as those in the Gulf of Lion, can also significantly influence the flow.

The analysis of the control vector elements shows that corrections to the open boundary conditions have the largest impact on CC transport changes, followed by initial conditions and atmospheric forcing. During periods of major ECC modifications, barotropic adjustments to the BC act continuously throughout the DA window generating positive or negative free surface gradients in the state increment that are associated with increases or decreases in CC transport. Meanwhile, DA ensures the consistency of the model with observations throughout the domain. When atmospheric forcing adjustment drives the CC transport variations the associated heat flux increment are in agreement with the patterns observed by Astraldi and Gasparini (1992).

Future work will extend the analysis both backward and forward in time to evaluate the impact of assimilated observations on the interannual variability of CC transport. Additionally, further attention will be dedicated to investigating the dynamics of the WCC and its role in the broader regional circulation.

This work represents an additional step toward an operational 4D-Var data assimilation forecasting system for the North-western Mediterranean sea, given the significant effect the employed observations have in constraining the circulation of the area for both the analysis and the forecast stages.

*Data availability.* The data that support the findings of this study are available from the corresponding author, M.B., upon reasonable request.

*Author contributions.* **M.B.**: conceptualization, formal analysis, investigation, methodology, software, supervision, validation, visualization, writing – original draft, writing – review and editing. **A.M.M.**: conceptualization, methodology, writing – review and editing. **R.S.**: conceptualization, writing – review and editing. **C.B.**: conceptualization, resources. **K.S.**: data curation, funding acquisition, writing – review and editing. **M.B.**: data curation, funding acquisition. **M.G.M.**: conceptualization, data curation, funding acquisition, resources, writing – review and editing.

*Competing interests.* The authors declare that they have no conflict of interest.

*Acknowledgements.* This research was funded by the European Union—NextGenerationEU and the Ministry of University and Research (MUR), National Recovery and Resilience Plan (NRRP), Mission 4, Component 2, Investment 1.5, project "RAISE—Robotics and AI for Socio-economic Empowerment" (ECS00000035). **M.B.** was fully supported by the the European Union—NextGenerationEU and the Ministry of University and Research (MUR), National Recovery and Resilience Plan (NRRP), Mission 4, Component 2, Investment 1.5, project "RAISE—Robotics and AI for Socio-economic Empowerment" (ECS00000035) project. **R.S.** and **M.G.M.** were partially supported by the EU - Next Generation Mission 4, Component 2 - CUP B53C22002150006 - IR0000032 – ITINERIS (Italian Integrated Environmental Research Infrastructures System) Project. The authors acknowledge partial support from the European project JERICO-S3 (H2020 grant agreement No 871153) for the data collection at the Corsica Channel mooring site.

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
