# Peer review of "Impact of Assimilated Observations on the Corsica Channel Transport in a 4D-Var System for the Northwestern Mediterranean Sea"

_EGUsphere, 2025_

## Referee Comment (RC1)

*Review for Ocean Science*
*Manuscript n°2025-3806*

**A 4D-Var Data Assimilation System for the North-Western Mediterranean Sea and its Impact on the Corsica Channel Transport**

Michele Bendoni, Andrew M. Moore, Roberta Sciascia, Carlo Brandini, Katrin Schroeder, Mireno Borghini, and Marcello Gatimu Magaldi

https://doi.org/10.5194/egusphere-2025-3806

**General comment**

The research conducted is based on studying the impact of each set of observations, assimilated into a ROMS configuration of the northwestern Mediterranean using the 4D-Var assimilation scheme, on the transport that characterizes the East Corsica Current (ECC) in the Corsica Channel (CC). This study follows on from that of Bendoni et al. (2023) who constructed the configuration and studied the impact of two sets of HFR data plus satellite SST data on the representation of the Northern Current transport. Major improvments enhance the value of the present article. The assimilated observation data has substantially increased (by approximately +40%) thanks to the addition of three new types of data (in situ ADCP mooring in the CC, in situ temperature and salinity profiles and SLA). Although one radar (Toulon) was removed from the assimilated sets, the detection area of the other (Ligurian) was doubled, as it is closer to the CC. The study period was also extended (four times) to cover the entire year 2022, which made it possible to evaluate the correct reproduction of the seasonal variability of ECC transport. In addition, numerous impressive indicators (Tables 1 and 2, Figures 3, 4, 7, 8, 9, 10) yielded robust results in accurately measuring the specific contributions of each observation to the targeted representation of ECC transport. It was demonstrated that the CC mooring was crucial in the DA procedure, while DA of satellite SLA was proved to be almost useless in the ECC representation. Furthermore, the authors showed that observations did not contribute equally across seasons and even show some opposite contributions that refine the vertical distribution of the transport in the CC transect, and that distant observations contribute to the transport refinement.

However, several issues remain to qualify the quality of the article. One-third of the text and half of the figures are grouped together in a single subsection of the results, with multiple references linking the figures in the twenty or so paragraphs of the subsection, to which are added inserted discussions, making it difficult to read. The objective of the study is to « provide insight into dominant dynamic scales » while some major features of the domain and period are not taken in account in the interpretation of the « different observations to transport increments through the CC », such as the upwelling system north of the Gulf of Lion, which is indirectly linked to the Bonifacio Gyre by the same atmospheric driving force (the Mistral and Tramontane winds), and precisely, the involvement of the Bonifacio gyre in modulating the variability of the ECC transport, or the intense and widespread marine heatwave that affected the entire Western Mediterranean from mid-May to the end of September in 2022. The study focuses solely on the western part of the CC, without considering the eastern part of the strait, which is shallower and sometimes features an eddy surrounding the island of Capraia during the summer. I therefore recommend that this article be published only after major revisions have been made.

*General remarks*

Section 3.2 is substantial ; it must be divided or subdivided. A discussion section is also highly recommended.

Please use the same nomenclature for section and subsection titles (with or without capital letters). Remove acronyms.

In general, figures could be more understandable without having to read their captions or the text (dates in the panels for example).

Please ensure compliance with the author guidelines.

*Specific remarks*

Please modify the title, it is too close to Bendoni et al. (2023) and does not adequately reflect the quality of your work. You could consider words such as « assessment » or « refinement » or put « 4D-Var DA system » latter in.

1. Introduction

Line 29. as instead of ad.

L 30. « MPA » is not used latter, not necessary.

L 31-32. The WMDW is also present in the Liguro-Provençal basin, as well as the WIW between AW and EIW. And the TIW is comprised in the ECC (Napolitano et al., 2019).

L 32-34. The cyclonic circulation is called the Northern Gyre and Liguro-Provençal basin has more currents than the NC, WCC and ECC, such as the Balearic current for instance. NC, WCC and ECC are focused on the Ligurian Sea.

L 34. LPC is not used after. And the NC also pursue along the Catalan coasts.

L 38-39. The eddy surrounding the island of Capraia is an important process in the CC (e.g. Iacono and Napolitano, 2020 among others). The Bonifacio Gyre is also important in the feeding and shaping of the ECC. The main circulation of the Tyrrhenian basin must also be mentioned.

L 40. « influences Western Mediterranean » ... « and the WIW and WMDW formation processes » ...

L 41. Could you add also a more recent reference for the importance of the CC in the biological connectivity between the Tyrrhenian and the Ligurian ?

L 45-57. An example of assimilation of mooring data that improved transport in a strait elsewhere ? In order to support your research. As well as for CORA and EMSO.

L 50-54. What kind of observations ?

L 70-72. Rewrite according to the changes.

2. ROMS 4D-Var Da System for the *North-Western Mediterranean*

2.1. Model setup

L 79. There are no explanations concerning Vtransform, Vstretching, ThetaS and ThetaB. A concise reason for changing the values of Nu and KhiT ? And the previous ones, if that is the case ?

L 113-114. It is therefore best to choose one and use it consistently throughout the rest of the text.

L 112. Toulon HFR has not been used. A reason, more than its far location ? You already did in Bendoni et al. (2023) and it was crucial for the NC reproduction.

L 119. « detrended » or smoothed ? A trimestrial smoothing does not remove a seasonal cycle (an annual moving average?). Removing a daily smoothing does not remove the diurnal signal. I think you meant the opposite.

2.2. Assimilated observations

L 130. Is there a reference for JERICO's HFR data such as previously ?

L 134-135. Did you sub-sampled in the space too ?

L 148-149. You could specify the types of data that make up the CORA database. This should also supports your study.

L 151. The website link for Albatross data is not working.

L 155 vs L 158. You have specified all satellites for SLA data, but not for SST data. Please choose one or the other (it seems that SST data is more important than SLA data in this study).

L 163. « signals, such as in Zavala-Garay » ?

L 164-165 Parenthesis around years.

L 171-172. In addition to the mean, could you provide the standard deviation and the extremes along the period ? Figure 3 seem to show lacks in January and July (see the comment about the figure).

2.3. Quantifying impact of observations

L 190. You limited your work in the east to 9.8369°E. Why not 10.5°E? The entire ECC transport would have been calculated accurately, as is generally the case in the literature, and would therefore have allowed for a more accurate comparison with it. You would have been able to

detect the presence or absence of the Capraia eddy in the summer and evaluate the DA system's ability to reproduce it. I think you should explore the subject.

**3.1. Performance of DA system**

L 205. FR is not defined before in the text.

L 215. « are similar (especially for salinity, Figure 3d) »

L 218. Tcora also presents a regime shift in mid-May. This corresponds to the large MHW of 2022 (see McAdam et al. (2023) and Estournel et al. (2025)). This may be the origin of the changes in Fig. 2 (SST+SLA+CORA ~25 % of data). The episode of May is also visible Scora (evaporation) and SLA (thermo-steric evelation), with another episode in late-July (Scora and SLA again), also corresponding to the other maximum of 2022 MHW. SST episode of mid-March : due to a lack of data. What happened in mid-August (Uhfr, Vhfr and Scora) ? A Capraia eddy ? Water from the Arno ? Both ?

L 219. 52.5 % is an important improvement, linked to the MHW ?

L 227. Because of the seasonal varibility of v while u is not predominant in the ECC.

L 229-230. The hydrological characteristics of AW, WIW, and EIW are more sensitive to seasonal surface variability than deeper water masses, which assumes that the seasonal variability is reached thanks to the DA.

L 231. S is less sensitive to seasonal surface variability than T.

L 232. Rho rather means density in oceanography. r is acceptable for Pearson's correlation coefficient.

L 246-247. W1 is actually located in the Ligurian current, so it cannot be used to measure the WCC.

L 245-247. The general circulation switched from a Western Mediterranean gyre (Algero-Provencal + Tyrrhenian, see Millot and Taupier-Letage, 2005) towards the North-Western Mediterranean gyre (WCC+NC+BC, Northern Gyre or the North Gyre, see Garreau et al., 2018 or Barral et al., 2025) by constructing the ECC's inner seasonal cycle (Vara et al., 2019). The southeastern closure of the North Gyre has clearly been enhanced along the southern boundary of the domain thanks to the DA.

**3.2. Observation Impact on Corsica Channel Transport**

L 268. An explanation for the in situ T bias ? Is in situ S doing the opposite ? Is it because of the Bonifacio Gyre assimilation (Fig. 10e) ?

L 270. « CC transport (dI $\geq$ 0), and vice versa. », isn't it ?

L 275-276. Or even, what would support the most your point, in a reanalysis with 0.35 ±0.365 Sv in Barral et al. (2025), while they took into acount the eastern part of the Capraia Island. Otherwise,

L 277. Figure 6e) does not exist.

L 276-279. And the fact that you also neglicted the transport in the east of Capraia Island.

L 296-297. Could it be that observation contributes more and more to AD as it is located upstream? Could it then be that observation data for the current located further south than CoCM is relevant? Such as HFR and/or moorings?Could you clarify why TINO/SVIN were not used (Doronzo et al., 2025) ?

L 301-303. Please specify somewhere « Global » and « Datum ».

L 306-307. Is it possible that if Nobs increases then Datum decreases ? And if Nobs increases then Nw increases and consequently Global increases too ? If so, HFR, in situ T & S and SST are approximately following these assumption, whereas CoCM and SLA are not. Consequentely, CoCM can be described as crucial and SLA as of limited impact for ECC transport in this setup, precisely for the ECC transport measurement obviously, whereas they are both of the same order of magnitude for their proportion with respect to the total observations (4.1 vs 2.5 %, Fig. 1d).

L 311-313. The manuscript could note that the northward flow on the western shelf has increased on average ; it is a contrasting result.

L 328-329. If you choose the area-normalized unit (see comment about Figure 9) you could say here that your is an indicator of both at the same time.

L 331. « Furthermore … pattern ». The results appear to show the opposite; please confirm. I think that the WCC enhancement due to the DA could be here better characterized.

L 335. What velocity shows the most surface ADCP from CoCM ? SST data also contributes to the opposite impacts during this period and seem to help CoCM to counteract HFR, the SST being at the surface. Otherwise, plenty of days of summer show that SST counteracts HFR.

L 336. HFR has a shallow impact, such as bathymetry there.

L 338 + 342. These sentences need to be qualified with regard to the circulation : the anticyclonic structure could be interpreted as being the Bonifacio Gyre whereas the northern part of the gyre's dipole is supposed to be cyclonic. A vertically integrated transport over two layers where currents counteracts is difficult to understand. From mid-November to December, IC and AC exert an important influence, in a transition zone between a southward and a strongly northward current. Maybe winds first intensified the Bonifacio gyre and, once the winds calmed down, the gyre weakened, such as the gyre favored the southward flow, while its weakening allowed the northward current to strengthen. If true, this proves that DA generates the ECC modulation by the gyre, without the latter being present in the domain (by a correction of the BC as explained in lines 343-345) and is a valuable result.

L 352-365. The freshwater influence regions (RoFi) of the Rhône and Arno rivers do not correspond to the yellow areas shown in Fig. 10c).

The RoFi of the Rhône does not extend to 6°E; it sometimes reaches 5°E, but less than a quarter of the time during the year (see Fraysse et al., 2014). The large yellow pattern located in the Gulf of Lion corresponds to the upwelling system formed by the Mistral wind and the downwelling system formed by easterly winds in the region (Odic et al., 2022). This coincides with your panels : the locations of upwellings and downwellings (Fig. 10c and e) during the subsurface MHW of 2022 in the Gulf of Lion (see Fig. 5a and 6b in Estournel et al., 2025).

The DA therefore both used the upwelling and downwelling systems, whether this was due to the DA link to the wind that builds the Bonifacio Gyre (that modulates the ECC especially during summer) or the subsurface MHW that may have weaken the ECC even more than a typical summer weakening of the ECC (by an attuation of the geostrophy due to stratification), or both of them at once.

One aspect is particularly interesting here: south-east of Corsica has a high RMSg in Fig. 10e), which does not appear in either Fig. 10c) or Fig. 10f) (or very weakly compared to the Rhône's RoFi importance). This may highlight the importance in situ T data from the intermediate layer there (no SST signal, no coastal signal, no salinity signal) and thus, either signs the MHWs or help to identify which data may have led to the BC improvment for the Bonifacio Gyre, or even both. A time series (such as done for SLA and not shown) may help here.

The RoFi of the Arno river does not extend beyond the north of La Spezia, and sometimes even goes south. It is rather localized, very coastal, on the surface, between Pisa and La Spezia (Vignudelli et al., 2004 ; Lapucci et al., 2012 ; Poulain et al., 2020). In this area, RMSg of SST is lower than offshore, whereas the river's freshwater is colder than the sea on average, even in winter (Vignudelli et al., 2004), and the Rhône issue showed that SST is not the best indicator for a RoFi. This yellow pattern of high SST RMSg that is located between Portofino and La Spezia more corresponds to the HFR RMSg distribution seen in Fig. 10a), which is linked to the ECC circulation in the CC. It seems that SST impact is here linked to HFR impact. Note the circulation of Fig. 9d) and the results of Bendoni et al. (2023) where they concluded that « The impact of the SST is significant and acts to further correct the velocity distribution of the increment induced by the HFRs observations ».

However, the RoFi of the Rhône does indeed have an influence on the DA. This phenomenon is illustrated in Figure 10f) (instead of 10c), which appears to delineate the significance of low salinities in the DA. This signal from low salinities does not reach Marseille, which corresponds to the findings of Fraysse et al. (2014). It should be noted that the aforementioned assistance is not applicable to the RoFi of the Arno river, due to the absence of in situ S data in that particular location. The relative importance of the DA of the Rhône river's RoFi for the ECC transport refinement is a subject of considerable interest, given that the RoFi is located downstream and is separated by the NC as a dynamic barrier. Are precipitations and the substantial discharges of the Rhône river seen as indicators for another process that controls the ECC ? Are the Arno southward extensions linked to the high precipitations ? It appears that in situ S data is also of significance in the western region of the Island of Elba (south of CoCM). Are these two locations of high in situ S RMSg correlated in some episodes alogn the period ? Does the Rhône exert a controlling influence over the BCs in the west ? In consideration of these DA results, which evince a counterintuitive phenomenon, it becomes imperative to propose a hypothesis or explanation for the observed occurrence of remote control.

4. Conclusion

L 429. « Corrections due to »

**Figures and Tables**

Please remove « 2022 » from times axes in figures 2, 3, 6, 8 and 10, and put all the months.

Fig. 1a. The 2022 surface circulation of the FR is redundant with figure 5a (even if not at the surface). The 'real' known average circulation of 2022 should be shown for the reader to have a reference. You could use the 2022 mean surface (or 0-200m, maybe useful for Fig.5) circulation from the reanalysis of Escudier et al., (2021).
Exchange 'b)' with 'c)', not the panel, only the letter.
Here you provide general information about observations, so another issue here is linked with figure 3 :

Fig. 3. Is the entire number of observation highly variable all along the 2022 year ? With a look at all grey bars, it seems that the lack of data, that is present in a) and b) during January and July, is balanced by panel e) ? Please confirm or show me or in the paper the entire time series of number of data.

Fig. 4. W1 data is lower than 0,0 %. Can you provide a reason ? Overall, W1 data is not mentionned while being close to the ECC.

Fig. 8d. Change FC to AC because you used FC for forecast (and adapt in the text).

Tab. 2. There no column for Nw. Can you provide a reason ? It is linked to Datum.

Fig. 9. Please label panels a-b « CoCM period (01 Jan–09 Apr) », panels c-d « HFR period (22 Apr-11 Aug) », and panels e-f « CoCM-HFR opposing period (18 Nov – 23 Dec) ». Please add in the caption « Periods are choosen with regard to Figure 8c » and adapt L 320-322.
'Sv*' is unclear. Please clarify whether it denotes Sverdrups divided by the horizontal area. If yes, use the unit 'Sv.m$^{-2}$' and state that it represents depth-integrated transport per horizontal unit area, equivalent to the column-average velocity in the same units. If it is not area-normalized, please specify the denominator explicitly (e.g., 'per water column' or 'per grid point' instead of 'per grid cell'). For clarity and consistency, I recommend the area-normalized option or another explicit unit, used consistently across the text and figures. This clarification would strengthen the presentation.

Fig. 12. I think that the caption is linked to an old figure.

**References**

Please put the DOIs, which is specified in the guide for authors (https://ocean-science.net/submission.html#references)

The references' section needs data references to be accompanied by the notation [dataset] (author's guide), such as those in lines 24 to 28, and lines 147 to 156.

LI data from offshore the Gulf of Lion is be cited with Houpert but should be also cited as the way that the website asked at first.

*Cited references here in this review that are not in the preprint :*

Napolitano et al., 2019, «The Tyrrhenian Intermediate Water...», PiO.
Iacono and Napolitano, 2020, « Aspects of the summer... », DSR P1.
McAdam et al., 2024, « Forecasting the Mediterranean... », SotP.
Estournel et al., 2025, «Extreme sensitivity... », Ocean Science.
Millot and Taupier-Letage, 2005, « Circulation in the Meditteranean Sea », book.
Garreau et al., 2018, « High-resolution Observations... », JGR Oceans.
Barral et al., 2025, « Assessment of the Water Mass Dynamics... », JGR Oceans.
Vara et al., 2019, « Role of atmosperic... », Ocean Modelling.
Doronzo et al., 2025, « Validating HF Radar... », Remote Sensing.
Fraysse et al., 2014, « Intrusion of Rhone River diluted ... », JGR Oceans.
Odic et al., 2022, « Sporadic wind-driven upwellin/downwelling... », CSR.
Vignudelli et al., 2004, « Distributions of dissolved... », ECS.
Lapucci et al., 2012, « Evaluation of empirical... », JARS.
Poulain et al., 2020, « On the dynamics in the ... », CSR.

---

## Referee Comment (RC2)

**General observations:**

The manuscript evaluates the impacts of different observation types on the Corsica Channel transport in a simulation of the NW Mediterranean Sea using the ROMS model with 4DVar data assimilation scheme. It follows a well-established methodology that have been applied many times to different regions around the globe. It represents an interesting contribution, that can be helpful in setting up a forecast system and to define future investments in the regional ocean observing system.

I feel the text would benefit from a major revision to strengthen its conclusions and improve the readability and figures. In particular, the discussion is fragmented and the figures are inconsistent in use of axes and colorbar ranges, making them hard to compare.

The conclusions summarize well the main results, but lack a closing statement that clearly defines the contribution of this paper and the expected importance in the larger picture.

**Detailed comments:**

**Introduction:**

Highlight the differences of your simulation to the Mediterranean Sea Physics Reanalysis, apart from resolution. What difference does your higher resolution makes? Why is this necessary?

Introduction lacks flow. In special the transition between the examples of 4DVar systems and the oceanography of the north-western Mediterranean Sea at lines 28-29.

Lines 29-35 – Will become more readable if broken up into smaller ones.

Lines 58-60 - This sentence is somewhat misleading, since modelling studies without DA can't by definition assess the impacts of DA. I recommend reviewing and potentially removing the reference to the non-DA simulations here.

Lines 64-65 - I don't understand how the observation impacts can be used to evaluate the dynamical scales. Could you, please, explain.

Lines 66-69 – It gives the impression that there are new observations assimilated that represent the "improvement" in relation to Bendoni et al. (2023). However, the first paragraph of session 2.1 gives a different idea. Therefore, this paragraph could use some rephrasing for more clarity.

**Session 2.1:**

Line 76 – Please specify the version of ROMS you used.

Line 111 – Can you justify your choice for the number of outer/inner loops and assimilation window length? Where there any sensitivity experiments?

Lines 112-114 – This interchangeable use of the terms "forecast" and "background" is confusing. Background is the accepted term used in data assimilation literature, while "forecast" is reserved to the "free run" initialized from an analysis. Please, review this and correct it through the text.

Moreover, I don't understand how this FR was run. Did you do a forecast after each analysis? Why not use your previous 4DVar analysis as initial conditions for the next assimilation window? Did you consider overlapping analyses? Please, clarify how this was run and the rationale behind it.

**Session 2.2:**

You give no explanation for how your observation errors were defined. The values look like what I would expect for instrument error, where representativeness errors would be expected. In addition, there is no spatial structure to the errors. Could you, please, justify your choices?

Lines 127-135 – The 3-hour low pass filter of the HFR velocities would still contain a tidal signal. However, you did not mention if and/or how you included tides in your simulation. Although the tidal signal is small in this area, this inconsistency can impact a "strong-constraint" DA scheme where all main physical processes are in principle included. Could you, please, elaborate how you deal with this?

Lines 145-168 – Please, break into separate paragraphs by observation type. This will help make it clear the different procedures applied to each observation.

Line 146 – Please, add the reference for CORA.

Lines 154-155 — I am curious why you chose to assimilate a SST product with a coarser resolution in relation to our model grid. Could you, please, justify your choice and the impact of errors that are carried from the SST product?

Lines 160-161 – Is the bias correction for the ADT compatible with the Reanalysis used for boundary conditions?

Lines 164-169 – Is the background QC applied to all the observations, or only the ADT? If only the ADT, why?

**Section 2.3:**

Lines 189-191 – It could be helpful to show the transect position in a map. Figure 1 is a good place to add this. Also, could you provide the rationale for choosing this particular section?

**Section 3.1:**

Lines 199-204 – Since you use the ratio between the eigenvalues as a guiding metric to identify overfitting, it would be better to show it in Figure 2.

Line 205 – This is the first occurrence of "FR". Please, add the definition.

Line 217 – These should be positive values (a negative reduction would mean the simulations are getting worst).

Line 225 – Is the difference in the fit of u and v related to the variability of the velocity components? It would be helpful to show the observation standard deviations next to the RMSEs.

Lines 237-241 – Please, present the standard deviations for all observations to support your affirmation. This could be added to Table 1.

Lines 251-256 – All that the time series show is data was in fact assimilated.

**Session 3.2:**

There is a lot of jumping back and forth with the figures, which harms the readability of the paper. Please, consider rearranging your discussion.

Lines 260-262 — I disagree that that graphs show that there is no BIAS. A better way to show this is by plotting the bias in Figure 3. Moreover, the graph for in situ S does not look "evenly distributed" for innovations.

Lines 273-276 – Please explain how can the impacts on transport shown in Figure 7 be many times higher than the values and variability shown in Figure 8.

Lines 276-281 – So, the observational studies consider the velocity constant with longitude? And how do you explain the difference to the other model studies? t is not good enough to say additional analysis could be done. Please, explain this better.

Lines 282-287 – What explains this variability and the differences in the assimilating model?

Lines 304-308 – Please, explain why the impact per observation is bigger in situ T and S in relation to HFR.

Lines 309-313 – This calls for a look at the vertical structure (transect) of the increments. You could link to the discussion of Figure 9.

Lines 339-347 – You kind of hint at the potential issue here: The placement of your southern boundary. Given the importance, please expand your discussion.

Line 348 – Here is a jump in your discussion. You should add some explanation and refer to session 2.3.

Lines 352-355 – This again shows the weight given to your southern boundary, on top of potential issues on the way the river flux is added to your simulation. How does the sea surface salinity maps look? Are the river plumes well represented?

Lines 380-395 – This is a very interesting analysis! As the heat flux reflects only what is happening at the surface, it would be good to see what is happening bellow. I believe the water column heat content or vertically integrated density would be helpful here.

**Session 4:**

I feel there is a missed opportunity here to discuss how effective the different observations are in constraining the simulation and the potential implications for forecasting and the "greater" western Mediterranean Sea – the usual goal of a assimilative system. You have lots of information that can be helpful in driving the evolution of the observing system.

A closing paragraph would do the job.

Table 1 – I am confused by your delta signals. While for RMSE a negative value represents improvement, the opposite seems to be truth for the correlation. I would rather see the actual values for the RMSE, correlation, and BIAS (instead of deltas). This would better make the point if your simulation is actually fitting well to the obs. And, please, be consistent with the +- signals.

Table 2 – Please, improve your caption to clearly indicate what you mean by "Global" and "Datum".

**Figures:**

Figure 1 - a) The "HFR" area is not clear. This can be resented by a polygon showing the coverage area. Also, what you present seem to be presenting is the mean stream function - this should be clear in your caption.

Here would be a great place to show the section chosen for the transport calculation used in your observation impact calculation.

- b) I feel the total observation density to not be very informative, and would be better plotted as a "pcolor" or "contourf". Instead, the distribution of each observation type can give a better idea of the coverage and expected impacts.
  - c) Why do you have an empty depth bin close to 2000m?
- Figure 2 Please, show the ratio between eigenvalues in sublot (b).
- Figure 3 Could you, please, use the same axis ranges for the number of observations in all your subplots? This will make it easier to compare.
- Figure 4 As for figure 3, please use the same axis limits for the number of observations. It would also be interesting to have the std of the (u,v) observations to be able to evaluate how much the variation of the fit by depth is relative to the variability of the currents.
- Figure 5 The reference to the subplots in your label is wrong. Always add the unit after the depth.
- Figure 6- Please, add the units for depth.
- Figure 7 Please, have use the same axis and colorbar for all your subplots. This figure is underexplored in the text, which makes me question if it is necessary.
- Figure 8 b) This is unnecessary. You don't seem to explore this figure enough in your text.

Subplots c) and d) arguably contain the most important results in your manuscript, and should be deeper explored in your text. Try to keep in mind what is the main point you want to make with your paper.

- Figure 9 Please, add your section position to the maps.
- Figure 10 Please, use the same limits for your map colorbars. The colormap you used is not helpful either. More contrast between values would make it easier to spot differences.

---

## Author Comment (AC1)

**Response to reviewer 1**

We thank the reviewer for the useful and inspiring comments. In the following, we first respond to the General observations and then we address point by point the General remarks and Specific remarks. For the sake of clarity, we enumerated them. Line numbers in the response to comments refer to the revised version of the manuscript unless otherwise stated.

**General comment**

*The research conducted is based on studying the impact of each set of observations, assimilated into a ROMS configuration of the northwestern Mediterranean using the 4D-Var assimilation scheme, on the transport that characterizes the East Corsica Current (ECC) in the Corsica Channel (CC). This study follows on from that of Bendoni et al. (2023) who constructed the configuration and studied the impact of two sets of HFR data plus satellite SST data on the representation of the Northern Current transport. Major improvements enhance the value of the present article. The assimilated observation data has substantially increased (by approximately +40%) thanks to the addition of three new types of data (in situ ADCP mooring in the CC, in situ temperature and salinity profiles and SLA). Although one radar (Toulon) was removed from the assimilated sets, the detection area of the other (Ligurian) was doubled, as it is closer to the CC.*

*The study period was also extended (four times) to cover the entire year 2022, which made it possible to evaluate the correct reproduction of the seasonal variability of ECC transport. In addition, numerous impressive indicators (Tables 1 and 2, Figures 3, 4, 7, 8, 9, 10) yielded robust results in accurately measuring the specific contributions of each observation to the targeted representation of ECC transport. It was demonstrated that the CC mooring was crucial in the DA procedure, while DA of satellite SLA was proved to be almost useless in the ECC representation.*

*Furthermore, the authors showed that observations did not contribute equally across seasons and even show some opposite contributions that refine the vertical distribution of the transport in the CC transect, and that distant observations contribute to the transport refinement.*

*However, several issues remain to qualify the quality of the article. One-third of the text and half of the figures are grouped together in a single subsection of the results, with multiple references linking the figures in the twenty or so paragraphs of the subsection, to which are added inserted discussions, making it difficult to read. The objective of the study is to « provide insight into dominant dynamic scales » while some major features of the domain and period are not taken in account in the interpretation of the « different observations to transport increments through the CC », such as the upwelling system north of the Gulf of Lion, which is indirectly linked to the Bonifacio Gyre by the same atmospheric driving force (the Mistral and Tramontane winds), and precisely, the involvement of the Bonifacio gyre in modulating the variability of the ECC transport, or the intense and widespread marine heatwave that affected the entire Western Mediterranean from mid-May to the end of September in 2022. The study focuses solely on the western part of the CC, without considering the eastern part of the strait, which is shallower and sometimes features an eddy surrounding the island of Capraia during the summer. I therefore recommend that this article be published only after major revisions have been made.*

The comments received proved to be very useful to improve the quality and readability of the work.

We rearranged the structure of the Results and Discussion section splitting into two different sections: 3. Results and 4. Discussion. Section 3 is, in turn, split into 3.1 Performance of the DA system, and 3.2 Observation impact, whereas Discussion is composed of 4.1 Changes in circulation and 4.2 Insights into DA mechanisms.

Although we are interested in the processes affecting the CC transport, the main subject of the paper is data assimilation (DA) and the physical processes that contribute to variations in the ECC are analyzed in light of the changes brought to them by DA.

In the discussion section we addressed the role of the Bonifacio Gyre in modulating the ECC transport and the way it is affected by the assimilation procedure despite not being fully contained in the computational domain. A comment is dedicated to the upwelling system in the Gulf of Lion with reference to its effect on the transport across the CC. The 2022 marine heatwave is mentioned with regard to the model skill in reproducing observed patterns. We verified the presence of the Capraia eddy during the summer season and mentioned this aspect in the text.

**General remarks**

*1) Section 3.2 is substantial ; it must be divided or subdivided. A discussion section is also highly recommended.*
We agree with the reviewer and we split the Results and discussion section into two sections: 3. Results and 4. Discussion. Section 3 is in turn split into 3.1 Performance of the DA system, and 3.2 Observation impact subsections, whereas Discussion is composed by 4.1 Changes in circulation and 4.2 Insights into DA mechanisms

*2) Please use the same nomenclature for section and subsection titles (with or without capital letters). Remove acronyms.*
Done

*3) In general, figures could be more understandable without having to read their captions or the text (dates in the panels for example).*
We added the dates in the panels of subplots in Figure 9 and we removed the year in the x axis ticks associated with the time series indicating all the months.

*4) Please ensure compliance with the author guidelines.*
Done

**Specific remarks**

*5) Please modify the title, it is too close to Bendoni et al. (2023) and does not adequately reflect the quality of your work. You could consider words such as « assessment » or « refinement » or put « 4D-Var DA system » latter in.*
We changed the title to: "Impact of Assimilated Observations on the Corsica Channel Transport in a 4D-Var System for the Northwestern Mediterranean Sea"

**1. Introduction**

*6) L 29. as instead of ad.*
Done

*7) L 30. « MPA » is not used latter, not necessary.*
Done

*8) L 31-32. The WMDW is also present in the Liguro-Provençal basin, as well as the WIW between AW and EIW. And the TIW is comprised in the ECC (Napolitano et al., 2019).*

*9) L 32-34. The cyclonic circulation is called the Northern Gyre and Liguro-Provençal basin has more currents than the NC, WCC and ECC, such as the Balearic current for instance. NC, WCC and ECC are focused on the Ligurian Sea.*

To reply to both comments, we modified the the text as follows:

"The north-western Mediterranean Sea, also known as Liguro-provençal Basin, is a crucial maritime region, hosting the Pelagos Sanctuary (Notarbartolo di Sciara et al., 2008) and several Marine Protected Areas (Francour et al., 2001). It is characterized by a cyclonic circulation (Northern Gyre) that involves Atlantic Water (AW) and modified Atlantic Water (mAW) at the surface, and Eastern Intermediate Water (EIW) and Tyrrenian Intermediate Water (TIW) at intermediate depths. Winter Intermediate Water (WIW) is mostly located at intermediate depth in the western part of the basin, whereas Western Mediterranean Deep Water (WMDW) occupy the deeper layers (Astraldi et al., 1994; Napolitano et al., 2019; Barral et al., 2024; Schroeder et al., 2024).

Around Corsica, the Eastern Corsica Current (ECC) and the Western Corsica Current (WCC) flow along opposite coasts and converge to form the Northern Current (NC, or Liguro-Provençal-Catalan Current), which flows cyclonically along the Italian and French coasts, up to the Catalan coast (Astraldi et al., 1990; Millot, 1999)."

Lines 29-38 updated manuscript.

*10) L 34. LPC is not used after. And the NC also pursue along the Catalan coasts.*

We removed the LPC acronym and added the information in the response to the previous comment.

*11) L 38-39. The eddy surrounding the island of Capraia is an important process in the CC (e.g. Iacono and Napolitano, 2020 among others). The Bonifacio Gyre is also important in the feeding and shaping of the ECC. The main circulation of the Tyrrhenian basin must also be mentioned.*

We expanded the text to include such information:

"The northward transport across the CC shows a marked seasonal cycle, with higher values during winter and spring and a net reduction in summer and autumn (Astraldi and Gasparini, 1992), occasionally reversing direction (Sciascia et al., 2019). This modulation aligns to the seasonal dynamics of the Tyrrhenian Sea. In winter it is characterized by a large-scale cyclonic circulation, when both surface and intermediate waters flow along the Italian coast and bifurcate, one part reaching the CC, and the other veering southward to join a semi-permanent cyclonic structure close to the Bonifacio Strait called Bonifacio Gyre. In summer, most of the water mass is recirculated within the Tyrrhenian basin with little outflow toward the Ligurian Sea (Artale et al., 1994; Astraldi and Gasparini, 1994). In addition, the modulation of transport through the CC can be affected by the presence of a recurrent anticyclonic structure, peculiar of the summer season and located in the channel area, known as Ligurian Anticyclone (LA) (Ciuffardi et al., 2016; Iacono and Napolitano, 2020)."

Lines 42-50 updated manuscript.

*12) L 40. « influences Western Mediterranean » … « and the WIW and WMDW formation processes »*

Modified as follows:

"The CC significantly influences Western Mediterranean dynamics by affecting the Ligurian Current and the WIW and WMDW formation processes (Schroeder et al., 2020)..."

Lines 51-52 updated manuscript.

*13) L 41. Could you add also a more recent reference for the importance of the CC in the biological connectivity between the Tyrrhenian and the Ligurian ?*

We did not find a recent paper directly relating the transport across the CC and the biological connectivity between the Tyrrhenian and Ligurian Seas, except for the one reported below:

- Aliani, S., Meloni, R., 1999. Dispersal strategies of benthic species and water current variability in the Corsica channel (Western Mediterranean). Sci. Mar. 63 (2), 137–145. https://doi.org/10.3989/scimar.1999.63n2137.

*14) L 45-57. An example of assimilation of mooring data that improved transport in a strait elsewhere ? In order to support your research.*

We added the following reference:

- Panteleev, Gleb, et al. "An inverse modeling study of circulation in the Eastern Bering Sea during 2007–2010." Journal of Geophysical Research: Oceans 121.6 (2016): 3970-3989.

*15) As well as for CORA and EMSO.*

CORA data are assimilated into the following products:

- Jean-Michel, Lellouche, et al. "The Copernicus global 1/12 oceanic and sea ice GLORYS12 reanalysis." Frontiers in Earth Science 9 (2021): 698876.

- Copernicus Climate Change Service, Climate Data Store. "ORAS5 global ocean reanalysis monthly data from 1958 to present." Copernic. Clim. Change Serv.(C3S) Clim. Data Store (CDS) (2021).

We were unable to confirm whether any other study described the assimilation of EMSO data.

We changed the text as follows:

"In situ data were employed by Hernandez-Lasheras and Mourre (2018) to compare model results after assimilating glider and CTD (conductivity-temperature-depth) observations, and by Aydogdu et al. (2025) to examine the impact of the assimilation of autonomous glider data comparing outputs from three different models with partially overlapping domains. In situ observations from specific datasets, such as CORA (Szekely et al., 2025), have been successfully assimilated into global and regional reanalysis (Jean-Michel et al., 2021; Zuo et al., 2018). Moreover, the assimilation of velocity data from mooring has proven to be a key ingredient to improve the transport, as shown by Panteleev et al. (2016) with a reanalysis of the Eastern Bering Sea circulation using a two-way nested 4D-Var system."

Lines 66-72 updated manuscript.

*16) L 50-54. What kind of observations ?*

We mentioned it in the text:

"Marmain et al. (2014) used HFR observations to adjust atmospheric forcing…"

Line 61 updated manuscript.

*17) L 70-72. Rewrite according to the changes.*

Done, now the text reads:

"The DA configuration and the observation impact methodology are described in Section 2. The effect of DA on circulation with respect to a non-assimilative model run is addressed in Section 3, focusing both on the agreement with observations (Subsection 3.1), and on the result of the observation impact methodology applied to the Corsica Channel transport (Section 3.2). The discussion about the changes induced by DA on the system and the related mechanisms are reported in Section 4.1 and 4.2, respectively. The last section is dedicated to conclusions and outlooks."

Lines 91-94 updated manuscript.

**2. ROMS 4D-Var Da System for the North-Western Mediterranean**

**2.1. Model setup**

*18) L 79. There are no explanations concerning Vtransform, Vstretching, ThetaS and ThetaB.*

These are specific parameters which affect the vertical sigma layers discretization. Vtransform and Vstretching decide which transformation to be used to pass from physical to dimensionless coordinates, whereas $\theta_s$ and $\theta_b$ control the degree of stretching for the surface and bottom layers, respectively. We slightly modified the text as follows:

"the following model parameters are adopted for the characterization of the vertical discretization of the grid: $V_{transform}$ = 2, $V_{stretching}$ = 4, $\theta_s$ = 7, $\theta_b$ = 2, and for the horizontal eddy viscosity v = 5 $m^2$ /s and diffusivity $\kappa_T$ = 1 $m^2$ /s."
Lines 100-102 updated manuscript.

*19) A concise reason for changing the values of Nu and KhiT ? And the previous ones, if that is the case ?*
We made the changes reducing them since the numerical scheme is already diffusive and we intended to reduce a little the diffusivity without making a large change with respect to the previous paper (we think the reviewer refers to Bendoni et al., 2023).

*20) L 113-114. It is therefore best to choose one and use it consistently throughout the rest of the text.*
We can understand the reviewer's request but in data assimilation the two terms can refer to the same simulation and, based on the context, they have a different sense. In our framework, we use the same length for the forecast and for the assimilation window. We run the (n)th forecast for 3 days starting from the last time step of the previous (n-1)th analysis. Such a forecast is indeed the background for the n(th) analysis cycle. When we refer, for example, to the run about which the nonlinear model is linearized, we use the term background. When, on the contrary, we refer to the skill and performance of the run starting from the analysis in an analysis-forecast sequence, we use the term forecast.
This is a case where background and forecast overlap, but they can differ (e.g. in case the assimilation window is 3 days and the subsequent forecast is 7 days). We believe it is important to keep them separated based on the context of reference.

*21) L 112. Toulon HFR has not been used. A reason, more than its far location ? You already did in Bendoni et al. (2023) and it was crucial for the NC reproduction.*
The reason is that, unfortunately, those data are not available for the year 2022, if not as daily averaged values.

*22) L 119. « detrended » or smoothed ? A trimestrial smoothing does not remove a seasonal cycle (an annual moving average?). Removing a daily smoothing does not remove the diurnal signal. I think you meant the opposite.*
Our intent is to reduce the signal variability to avoid the standard deviation being too large and giving rise to an excessively large value for the background error covariance and, consequently, to an exaggerated correction to the solution. To avoid misunderstanding we rephrased as follows:
"To avoid too large standard deviation values for temperature and heat flux, we removed from the original time series the moving average at quarterly timescale, and then the moving average at daily timescale for each cell of the domain."
Lines 146-148 updated manuscript.

**2.2. Assimilated observations**

*23) L 130. Is there a reference for JERICO's HFR data such as previously ?*
We provided a link to the data and one for the JERICO research infrastructure.
Lines 157-158 updated manuscript.

*24) L 134-135. Did you sub-sampled in the space too ?*
We averaged them over a 6km square grid but we did not sub-sampled.

*25) L 148-149. You could specify the types of data that make up the CORA database. This should also support your study.*

We modified the text as follows:

"They are part of the Global Ocean CORA In situ Observations, collected mainly by Argo Floats, XBT, CTD, XCTD and moorings, and we refer to them as $T_{CORA}$ and $S_{CORA}$…"

Lines 177-178 updated manuscript.

*26) L 151. The website link for Albatross data is not working.*

It is strange since the link

https://www.seanoe.org/data/00720/83244/

brings to the ALBATROSS web page.

*27) L 155 vs L 158. You have specified all satellites for SLA data, but not for SST data. Please choose one or the other (it seems that SST data is more important than SLA data in this study).*

The problem is that the gridded SST product is directly available from a single source, while the SLA tracks are accessible split by satellite. That is the reason why we reported such information for the SLA. We removed the satellite list for the SLA to be consistent with the SST description. This information is available on the User Manual and Quality Information Document provided by the CMEMS platform.

We also corrected the ID of the product SST_MED_PHY_L3S_MY_010_042 instead of SST_MED_SST_L3S_NRT_OBSERVATIONS_010_012

*28) L 163. « signals, such as in Zavala-Garay » ?*

Line 194 updated manuscript.

*29) L 164-165 Parenthesis around years.*

Line 196 updated manuscript.

*30) L 171-172. In addition to the mean, could you provide the standard deviation and the extremes along the period ? Figure 3 seems to show lacks in January and July (see the comment about the figure).*

The standard deviation of observation number per assimilation cycle has been added to the text, together with the maximum/minimum number of observations for a single DA window.

"The amount of observations per assimilation window throughout the year is reported in Figure 1e. In total, approximately $4.5 \cdot 10^6$ observations were assimilated during 2022, with a mean of $37.12 \cdot 10^3$ observations per assimilation cycle, a maximum of $49.15 \cdot 10^3$, a minimum of $16.29 \cdot 10^3$ and a standard deviation equal to $5.82 \cdot 10^3$. Further temporal and depth-dependent distributions by data source are presented in Figures 3 and 4."

Lines 202-205 updated manuscript.

**2.3. Quantifying impact of observations**

*31) L 190. You limited your work in the east to 9.8369°E. Why not 10.5°E? The entire ECC transport would have been calculated accurately, as is generally the case in the literature, and would therefore have allowed for a more accurate comparison with it. You would have been able to detect the presence or absence of the Capraia eddy in the summer and evaluate the DA system's ability to reproduce it. I think you should explore the subject.*

In this work we specifically focus on the dynamic related to the Corsica Channel. We are aware that the transport across the transect spanning the area between the Island of Capraia and the Italian coast has an effect on the large-scale circulation of the area but we believe it is not significant as it is

the CC transport. To our knowledge, most of the literature cited refers to the transport across the Corsica Channel and the transport computed from observations refers to the deepest part of the channel where the CoCM is located. The only three studies which compute the transport across the whole transect (Corsica-Italian coast) are those by De la Vara et al. (2019) and Barral et al. (2024), and an observational study of Manzella (1985) using a mooring really close to the eastern coast of Capraia. However, we agree that the analysis of the transport associated to the whole transect deserves to be further studied and analyzed. In the discussion subsection 4.1 we have given more space to this topic:

"Recalling that the yearly averaged transport values that we obtained at the CC for FR, FC and AN are equal to 0.49, 0.31 and 0.28 Sv, respectively, these values are lower than previous estimates based on observations, ranging from 0.54 Sv to 0.71 Sv (Astraldi and Gasparini, 1992; Astraldi et al., 1994), and numerical models, such as 0.49 Sv in Sciascia et al. (2019) and 0.5 Sv in Béranger et al. (2005), the latter based on a multi-decadal time series. However, reanalysis by de la Vara et al. (2019) and Barral et al. (2024) obtained an average transport around 0.33-0.34 Sv and 0.35 ±0.365 Sv, respectively, considering the whole transect including the CC and the area from Capraia Island to the Italian coast.

Part of the discrepancy with observational estimates may stem from the assumption of negligible longitudinal variability in the northward velocity across the channel that is not fully supported by our model results (Figure 5e and Figure 8d, e) and was also observed by previous numerical studies (Sciascia et al., 2019). Interestingly, the two reanalyses of de La Vara et al. (2019) and Barral et al. (2024) obtained an annual average transport comparable to our result by considering the whole Tyrrhenian transect. These values lower than those from previous literature can be ascribed to a possible negative annual averaged transport for the area between the Island of Capraia and the Italian Coast, or to a more appropriate representation of the flow in the area.

The transport across the whole transect linking the Tyrrhenian and Ligurian Seas, and the interaction between the two sub-transects, also in light of basin-scale mass budgets, deserve further studies and analysis. Furthermore, additional analysis, particularly over a longer time period, will help determine whether the year 2022 reflects inter-annual variability (Vignudelli et al., 1999), or is the signal of a longer-term trend."

Lines 338-349 updated manuscript.

We checked the presence of the Capraia Eddy by inspecting the monthly averaged surface velocities from the FR, and AN. We did not find the presence of the eddy in summer seasons but, only for the analysis run an anticyclonic eddy is present north of Corsica for October.

We mentioned this in the discussion section after we analyzed the changes produced by DA to the overall circulation:

"We also looked for a peculiar pattern of the summer circulation in the area, that is the presence of the Ligurian Anticyclone (LA), also for its potential role in modulating the ECC (Ciuffardi et al., 2016; Iacono and Napolitano, 2020). The determination of monthly averaged surface velocity fields allowed us to exclude the presence of the LA in the summer season, for both FR and AN. However, in October, a cyclonic feature north of Corsica was present in the AN and not in the FR run (not shown). Significant current reversal episodes were not observed in the corresponding period, but the presence of the LA might have concurred to maintain the transport lower for the analysis than for the freerun (Figure 6a)."

Line 424-429 updated manuscript.

**3.1. Performance of DA system**

*32) L 205. FR is not defined before in the text.*
We modified the sentence at line 91 as follows: "The nonlinear model without assimilation (freerun, FR) was run from 2019 to 2022, after…"

Lines 116-117 updated manuscript.

*33) L 215. « are similar (especially for salinity, Figure 3d) »*
Corrected

*34) L 218. Tcora also presents a regime shift in mid-May. This corresponds to the large MHW of 2022 (see McAdam et al. (2023) and Estournel et al. (2025)). This may be the origin of the changes in Fig. 2 (SST+SLA+CORA ~25 % of data).*
We added this to the text:
"SST and $T_{CORA}$ show a notable RMSE increase for the FR, and less so for the FC, starting in mid-May, while $RMSE_{AN}$ tends to remain more stable (Figure 3c and e)."
Lines 252-253 updated manuscript.
We also reported the following in the discussion subsection 4.1:
"Looking at Figure 3c and e, a regime shift is observable in the RMSE for $T_{CORA}$ and SST, which is larger for both FR and FC with respect to AN, starting approximately from mid-May. We argue it can be attributed to the marine heat wave (MHW) that affected the Western Mediterranean sea from mid-May to the end of summer 2022 (McAdam et al., 2024; Estournel et al., 2025), despite the regime shift in RMSE is present until the end of the year for both $T_{CORA}$ and SST."
Lines 405-408 updated manuscript.

*35) The episode of May is also visible in Scora (evaporation) and SLA (thermo-steric elevation), with another episode in late-July (Scora and SLA again), also corresponding to the other maximum of 2022 MHW.*
We also added in continuity with the previous point:
"The effect of the MHW, although to a lesser extent, can be traced on the two RMSE peaks of $S_{CORA}$ in May and August, probably due to strong evaporation processes (Figure 3d)."
Lines 408-410 updated manuscript.
Concerning the SLA signal, we do not believe we can give the same interpretation related to the MHW, as for the other mentioned variables, to explain the RMSE pattern.

*36) SST episode of mid-March : due to a lack of data.*
Yes, it is possible that the few data available are not in agreement with the model.

*37) What happened in mid-August (Uhfr, Vhfr and Scora) ? A Capraia eddy ? Water from the Arno ? Both ?*
We looked at the surface quiver plot from the background solution associated to the assimilation window with the large RMSE in $v_{HFR}$ and $u_{HFR}$ (2022-08-11) and we did not find specific patterns that can explain the increase in the error. There was only a clear inertial oscillation pattern.

*38) L 219. 52.5 % is an important improvement, linked to the MHW ?*
It is likely that such a reduction is due to the fact that only the analysis can keep up with observations (see above point 34).

*39) L 227. Because of the seasonal variability of v while u is not predominant in the ECC.*
We reported the following text in the discussion subsection 4.1:
"If we focus on the physical processes affecting the performance of the DA system we can try to explain, for example, why the error reduction is more pronounced for the v than the u velocity component observed by the CoCM (Figure 4a, b, e and Table 1). Since the latter has a lower observed average magnitude (0.008 m/s) and standard deviation (0.022 m/s) with respect to the former (0.05 m/s and 0.09 m/s, respectively), the DA has more chances to correct v than u, also because we made the assumption that the background error covariance is proportional to the standard deviation of the

state variables. being the flow orographically constrained, the eastward velocity (u) cannot reach large values and possible small variations are more hardly caught by the model."
Lines 398-404 updated manuscript.

*40) L 229-230. The hydrological characteristics of AW, WIW, and EIW are more sensitive to seasonal surface variability than deeper water masses, which assumes that the seasonal variability is reached thanks to the DA.*
We mentioned this aspect in the discussion subsection 4.1:
"The difference in RMSE between AN and FR for $T_{EMSO}$ and $S_{EMSO}$, is remarkable for surface and intermediate water, while it reduces or disappears for deeper water (Figure 4c and d). It is likely explainable considering that AW, mAW and EIW are more sensitive than WMDW or TDW to seasonal changes that can be captured by DA."
Lines 411-414 updated manuscript.

*41) L 231. S is less sensitive to seasonal surface variability than T.*
Since S is less sensitive than T to seasonal variability, it is possible that the FR better reproduces the S instead of the T field. This larger RMSE initially associated with T can be related to seasonal variability that assimilation is not able to fully correct leading to a smaller RMSE reduction.

*42) L 232. Rho rather means density in oceanography. r is acceptable for Pearson's correlation Coefficient.*
Done, everywhere in the paper we substituted the symbol $\rho$ with *r* when referred to correlation.

*43) L 246-247. W1 is actually located in the Ligurian current, so it cannot be used to measure the WCC.*
We were not referring to the W1, we were considering the "L" mooring mentioned by Astraldi et al., (1990) and Astraldi and Gasparini (1992). However, having a better look at the documents, we revised our consideration about its location to measure the WCC. We removed the part after the citation (see next point 44).

*44) L 245-247. The general circulation switched from a Western Mediterranean gyre (Algero-Provencal + Tyrrhenian, see Millot and Taupier-Letage, 2005) towards the North-Western Mediterranean gyre (WCC+NC+BC, Northern Gyre or the North Gyre, see Garreau et al., 2018 or Barral et al., 2025) by constructing the ECC's inner seasonal cycle (Vara et al., 2019). The southeastern closure of the North Gyre has clearly been enhanced along the southern boundary of the domain thanks to the DA.*
We rephrased the sentence as follows:
"DA slightly intensifies the NC, primarily by strengthening the WCC, which is barely noticeable in the FR, while the ECC flow is reduced (Figure 9a and b). This pattern can be seen as a shift from a Western Mediterranean gyre, which principally involves the Tyrrhenian and Provencal basins, toward a North-Western Mediterranean gyre, which partially substitutes the Tyrrhenian inflow with waters coming directly from the Algerian basin. The ECC behavior aligns with previous studies showing its high variability (Astraldi et al., 1990; Astraldi and Gasparini, 1992), whereas the pronounced WCC signal was also found by Barral et al. (2024) and Ciuffardi et al. (2016)."
Lines 416-421 updated manuscript.

**3.2. Observation Impact on Corsica Channel Transport**

*45) L 268. An explanation for the in situ T bias ?*

We think the reviewer refers to the δI ($d_i \cdot g_i$ in the updated manuscript) associated to the in situ T, which shows an asymmetric behavior with a majority of negative increments (61% with respect to 39% of positive increments), associated to a slightly biased in situ d = y-H(x) values (47.1% negative, 52.9% positive). It is possible that the temperature profiles constrain the density structure of the water column leading to baroclinic pressure gradients which concur to reduce the transport.

*46) Is in situ S doing the opposite ?*
It seems in situ S is doing the opposite if we look at the amount of δI values. On the contrary, negative δI values are generally larger. Overall, we believe there is a substantial equivalence between positive and negative contribution of in situ S to the transport.

*47) Is it because of the Bonifacio Gyre assimilation (Fig. 10e) ?*
See response to point 70 below.

*48) L 270. « CC transport (dI > 0), and vice versa. », isn't it ?*
Corrected. Line 320 updated manuscript.

*49) L 275-276. Or even, what would support the most your point, in a reanalysis with 0.35 ±0.365 Sv in Barral et al. (2025), while they took into account the eastern part of the Capraia Island. Otherwise,*
We mentioned this. See the response to point 31 above.

*50) L 277. Figure 6e) does not exist.*
It is there. It is the subplot at the bottom right of Figure 5 (updated manuscript), showing the time averaged velocity distribution on the CC transect.

*51) L 276-279. And the fact that you also neglected the transport in the east of Capraia Island.*
See response to point 31 above.

*52) L 296-297. Could it be that observation contributes more and more to AD as it is located upstream?*
It can be possible, but it depends on several factors since it is the combination of the adjoint model and the error covariances which defines those portions of the domain affecting more the transport across the CC. (In responding, we assume the reviewer means data assimilation with "AD").

*53) Could it then be that observation data for the current located further south than CoCM is relevant?*
Yes, it is possible that observations covering the area southward of the CC are relevant for the channel dynamics. A useful tool to check it would the an adjoint sensitivity analysis specifically addressing the sensitivity of transport (d (Tr)/dz, where z here mean the control vector) with respect to a previous time window (Moore et al., 2009, An adjoint sensitivity analysis of the southern California Current; Zhan et al., 2018, Sensitivity studies of the Red Sea eddies using adjoint method).

*54) Such as HFR and/or moorings?*
The adjoint sensitivity would allow us to estimate the sensitivity with respect to the different variables and their space-time locations, thus u and v velocity components at the surface as measured by HFR and those along the water column, sampled by a mooring.
We discussed this point as follows at the beginning of the subsection 4.2 Insights into DA mechanisms:
"Observation impact can allow us to investigate in which ways DA modifies the model state with respect to a particular metric (transport across a section in our case) by optimally balancing between observations and background solution."
Lines 431-432 updated manuscript.

And the following:
"Developing more the analysis, we can even estimate the contribution of each observation to the transport increment through $\Delta Tr_{TL} = \mathbf{d}^T \cdot \mathbf{g}$. The presence of a seasonality in $\Delta Tr$ when we consider the observation typology (Figure 6c), that is roughly cold months controlled by CoCM while warm months until late autumn by HFR, could be explained by the corresponding seasonality in the CC transport. Small northward velocity values in warm months tend to produce small innovations $\mathbf{d}$ and, consequently, possible small $\mathbf{d}^T \cdot \mathbf{g}$. To better control the dynamic at the channel, DA relies on another source of observations that is HFR. In cold months, the opposite occurs and the role of CoCM observations is predominant."
Lines 439-444 updated manuscript.

*55) Could you clarify why TINO/SVIN were not used (Doronzo et al., 2025) ?*
We preferred to use the same HFR system employed for the Bendoni et al. (2023) paper and the TINO/SVIN system had a smaller temporal coverage for the year 2022. Since we intend to enlarge the temporal cover of the DA exercise, we also plan to make an analysis on the differences we obtain by assimilating separately the two HFR datasets, all other things being equal.

*56) L 301-303. Please specify somewhere « Global » and « Datum ».*
We modified the text, and changed the term "total" to "Global".
We also modified caption of Table 2 as follows:
"Root mean square impact for the different observation typologies, both globally (Global refers to the effect on transport of all observations belonging to a certain typology), and considering the average contribution of a single observation (Datum refers to the average effect on transport of a single observation belonging to a certain typology). The last column reports the total number of observations per type."
Table 2 caption updated manuscript.

*57) L 306-307. Is it possible that if Nobs increases then Datum decreases ?*
We believe it depends on both the type and location of observations, with respect to the chosen metric for the observation impact calculation. Datum can decrease as Nobs increases if some observations are redundant with respect to the information content of the whole dataset. A possible answer could be investigated if (e.g. for HFR data) we assimilate the same dataset at different levels of subsampling and compare the impacts.

*58) And if Nobs increases then Nw increases and consequently Global increases too ?*
This is our fault. There was a typo in the equation to calculate Global RMS impact and an error in the definition of $N_w$ that is simply the number of assimilation windows.

*59) If so, HFR, in situ T & S and SST are approximately following these assumptions, whereas CoCM and SLA are not. Consequently, CoCM can be described as crucial and SLA as of limited impact for ECC transport in this setup, precisely for the ECC transport measurement obviously, whereas they are both of the same order of magnitude for their proportion with respect to the total observations (4.1 vs 2.5 %, Fig. 1d).*
Despite our error (comment 58), the reasoning of the reviewer is correct.
We added the following in the subsection 4.1 Insights into DA mechanisms of the Discussion section:
"Table 2 reports the RMS transport increments considering each typology of observation (Global) and the single observation per typology (Datum). Apart from CoCM and SLA data, it seems that the impact of a single observation is inversely proportional to $N_{obs}$. It could be explained by considering that the larger the amount of observations, the higher the probability that some of them are redundant and do not provide additional information content. On the contrary, CoCM/SLA observations are characterized

by large/small Global and Datum RMS, given a comparable $N_{obs}$ (Table 2), showing that the former are crucial to constrain ECC, whereas the latter have a small effect for the present modeling setup."
Lines 433-438 updated manuscript.

*60) L 311-313. The manuscript could note that the northward flow on the western shelf has increased on average ; it is a contrasting result.*
We agree with the review and we mentioned this aspect also considering a similar pattern in Figure 9a (Figure 8a in the revised version of the manuscript). In the updated manuscript Figure 6 (Figure 5 in the revised version of the manuscript) belongs to the Results section. The following text is reported in the discussion subsection 4.1 and is modified to account for the reviewer's request:
"CoCM observations tightly constrain the flow near their location but with the side effect of accelerating the northward flow in the surface western part of the transect. In fact, the δv values, averaged over the whole year, are characterized by an overall reduction for the most part of the CC transect and a slight increase at the upper western flank (not shown). This may be the result of insufficient corrections to the boundary conditions, redirecting the volume transport to unconstrained portions of the domain (Figure 8a, Figure 5e)."
Lines 363-367 updated manuscript.

*61) L 328-329. If you choose the area-normalized unit (see comment about Figure 9) you could say here that your is an indicator of both at the same time.*
We modified the text, the Figure and the caption as suggested by the reviewer (see comment about Figure 9,points 86 and 86).

*62) L 331. « Furthermore … pattern ». The results appear to show the opposite; please confirm. I think that the WCC enhancement due to the DA could be here better characterized.*
Yes, there was an error, we substituted the term "weaker" with "stronger".
We gave space to the WCC at the end of the first subsection of the discussion "4.1 Changes in CC transport and overall circulation". See comment 44.

*63) L 335. What velocity shows the most surface ADCP from CoCM?*
The surface velocity (upper 50 m) measured by the CoCM mooring is not reliable and we did not consider them in the analysis. See lines 141-143.

*64) SST data also contributes to the opposite impacts during this period and seem to help CoCM to counteract HFR, the SST being at the surface. Otherwise, plenty of days of summer show that SST counteracts HFR.*
We modified the text as follows:
"Figure 8e reveals surface deceleration from HFR and subsurface acceleration from CoCM, with an additional northward acceleration provided by SST (Figure 6c) which is reasonably concentrated mainly at the surface."
Lines 382-383 updated manuscript.

*65) L 336. HFR has a shallow impact, such as bathymetry there.*
We apologize to the reviewer but we are not sure we correctly understood the meaning of the comment.

*66) L 338 + 342. These sentences need to be qualified with regard to the circulation : the anticyclonic structure could be interpreted as being the Bonifacio Gyre whereas the northern part of the gyre's dipole is supposed to be cyclonic. A vertically integrated transport over two layers where currents counteracts is difficult to understand.*

When we refer to the anticyclonic structure at line 338 we refer to the increment δTr(x,y) in transport all over the domain, that is what is added to the background to obtain the analysis.

*67) From mid-November to December, IC and AC exert an important influence, in a transition zone between a southward and a strongly northward current. Maybe winds first intensified the Bonifacio gyre and, once the winds calmed down, the gyre weakened, such as the gyre favored the southward flow, while its weakening allowed the northward current to strengthen. If true, this proves that DA generates the ECC modulation by the gyre, without the latter being present in the domain (by a correction of the BC as explained in lines 343-345) and is a valuable result.*

The anticyclonic increment in transport δTr(x,y) in case the CC transport increases can be explained by what the reviewer suggested. We modified the text as follows:

"This acceleration at the eastern flank of Corsica, and deceleration at the Tuscany coast, are present at the southern boundary, where an anticyclonic structure in the average transport increment is formed. We argue it is a modulation of the Bonifacio Gyre whose cyclonic structure, if weakened, tends to favour the transport in the CC for this particular current pattern. This happens in correspondence of a period (18 November - 23 December) where both IC and FC (other than BC) play a role in constraining the transport in the CC (Figure 6d) and it is indeed the atmospheric forcing that acts to modulate the intensity of the Bonifacio Gyre (Astraldi and Gasparini, 1994; Iacono and Napolitano, 2020).

This behaviour exemplifies a key feature of 4D-Var: the adjoint model $M^T_b$ uses the derivative $\partial I/\partial z$ of the CC transport to inform the system what modifications are required at the control vector to influence I during the 3 day assimilation window. For a more detailed explanation of the observation impact mechanism, see Levin et al. (2021)."

Lines 385-390 updated manuscript.

*68) L 352-365. The freshwater influence regions (RoFi) of the Rhône and Arno rivers do not correspond to the yellow areas shown in Fig. 10c).*
*The RoFi of the Rhône does not extend to 6°E; it sometimes reaches 5°E, but less than a quarter of the time during the year (see Fraysse et al., 2014). The large yellow pattern located in the Gulf of Lion corresponds to the upwelling system formed by the Mistral wind and the downwelling system formed by easterly winds in the region (Odic et al., 2022). This coincides with your panels : the locations of upwellings and downwellings (Fig. 10c and e) during the subsurface MHW of 2022 in the Gulf of Lion (see Fig. 5a and 6b in Estournel et al., 2025).*

The reviewer is right. We added this information and the associated reference. See also the text at point 75 below.

*69) The DA therefore both used the upwelling and downwelling systems, whether this was due to the DA link to the wind that builds the Bonifacio Gyre (that modulates the ECC especially during summer) or the subsurface MHW that may have weakened the ECC even more than a typical summer weakening of the ECC (by an attenuation of the geostrophy due to stratification), or both of them at once.*

We introduced and examined this aspect in the text, but did not find a satisfactory explanation for the underlying mechanism. This precise relationship between the upwelling/downwelling system and potential ECC variations would require a targeted investigation, focusing on specific episodes. See also the text at point 75 below.

*70) One aspect is particularly interesting here: south-east of Corsica has a high RMSg in Fig. 10e), which does not appear in either Fig. 10c) or Fig. 10f) (or very weakly compared to the Rhône's RoFi importance). This may highlight the importance in situ T data from the intermediate layer there (no SST signal, no coastal signal, no salinity signal) and thus, either signs the MHWs or help to identify*

*which data may have led to the BC improvement for the Bonifacio Gyre, or even both. A time series (such as done for SLA and not shown) may help here.*

We checked the targeted in situ T data (see figure below, first subplot) and we noticed that the large $g_i$ values correspond to the period end of November - beginning of December (see figure below, second subplot), and we can find traces of them in Figure 6c. However, their contribution is small and observations are located close to the surface.

[Figure]

*71) The RoFi of the Arno river does not extend beyond the north of La Spezia, and sometimes even goes south. It is rather localized, very coastal, on the surface, between Pisa and La Spezia (Vignudelli et al., 2004 ; Lapucci et al., 2012 ; Poulain et al., 2020). In this area, RMSg of SST is lower than offshore, whereas the river's freshwater is colder than the sea on average, even in winter (Vignudelli et al., 2004), and the Rhône issue showed that SST is not the best indicator for a RoFi. This yellow pattern of high SST RMSg that is located between Portofino and La Spezia more corresponds to the HFR RMSg distribution seen in Fig. 10a), which is linked to the ECC circulation in the CC. It seems that SST impact is here linked to HFR impact. Note the circulation of Fig. 9d) and the results of Bendoni et al. (2023) where they concluded that « The impact of the SST is significant and acts to further correct the velocity distribution of the increment induced by the HFRs observations ».*

It is correct what the reviewer is saying about the RoFI of the Arno river.
We removed the part on the ROFI associated with the Arno river and added a part in relation to this comment in the text at point 75 below.

*72) However, the RoFi of the Rhône does indeed have an influence on the DA. This phenomenon is illustrated in Figure 10f) (instead of 10c), which appears to delineate the significance of low salinities in the DA. This signal from low salinities does not reach Marseille, which corresponds to the findings of Fraysse et al. (2014). It should be noted that the aforementioned assistance is not applicable to the RoFi of the Arno river, due to the absence of in situ S data in that particular location. The relative importance of the DA of the Rhône river's RoFi for the ECC transport refinement is a subject of considerable interest, given that the RoFi is located downstream and is separated by the NC as a dynamic barrier. Are precipitations and the substantial discharges of the Rhône river seen as indicators for another process that controls the ECC ?*

We agree with the reviewer and added this in the text together with the suggested reference (see modifications to the manuscript at point 75 below).

*73) Are the Arno southward extensions linked to the high precipitations ?*

We believe the southward extension is more a question of visualization and is not necessarily linked to heavy precipitations. Indeed the extension reaches the latitude of the Gorgona Island and it is in line with what reported by Lapucci et al. (2012) and Vignudelli et al. (2004). We updated the colorbar for several figures, including Figure 10 leading to a more clear identification of the main patterns.

*74) It appears that in situ S data is also of significance in the western region of the Island of Elba (south of CoCM). Are these two locations of high in situ S RMSg correlated in some episodes along the period ?*

We performed for in situ S data the same procedure of point 70 above (see figure below where first subplot reports selected observations and the second one the time evolution of $g_i$). Also in this case observations are located close to the surface and $g_i$ assumes large values around mid-November. Data have a smaller effect on transport increment with respect to other observation typologies in the same period (Figure 6c).

[Figure]

*75) Does the Rhône exert a controlling influence over the BCs in the west ? In consideration of these DA results, which evince a counterintuitive phenomenon, it becomes imperative to propose a hypothesis or explanation for the observed occurrence of remote control.*

RMSg values come from the observation impact exercise performed on the CC transport. Furthermore, salinity observations at the Rhône mouth contribute only partially to the transport increment (around 8% if we consider the Global RMS impact), and it is not necessarily true that they directly exert a control on the west side of southern BC. A specific observation impact experiment or an adjoint sensitivity analysis on a transect located west of Corsica or close to the southern boundary would be needed to properly infer the control exerted by the Rhône river on it. We believe this will require additional and computationally expensive simulations and is beyond the scope of the present work.

In the following we report the modification adopted to respond to the above points, namely  from 68 to 75:

"The degree of sensitivity of the transport to SST and in situ T data is particularly high in the Gulf of Lion (GoL). Traces of the upwelling system induced by north-westerly winds (Odic et al., 2022) are detectable for both SST and in situ T (Figure 10c and e), whereas those associated to south-easterly winds at the western coast of the GoL (Odic et al., 2022) are mainly present in the map of in situ T RMSg (Figure 10e). Similarly, it appears that salinity variations in the region of freshwater influence (ROFI) of the Rhône river can contribute to the evolution of the ECC (Figure 10f). High sensitivities in the area may be partly explained by the background-error covariance matrix **B** that we assumed to be

proportional to the standard deviation of the freerun. For temperature, it displays large values at the surface in the eastern part of the GoL area and lower, but significant values, at the western part, not only at the surface (not shown). For salinity, large standard deviation values are restricted to the ROFI of the Rhône river (Fraysse et al., 2014) (not shown). However, the physical mechanisms through which upwelling/downwelling episodes or salinity variations in the GoL can influence CC transport, deserve and require further investigation.

Secondary hot-spots of RMSg for in situ T and S near the LI station and around the eastern coast of Corsica, especially for T, emphasize that distant observations can significantly affect CC transport (Figures 10e and f).

The large RMSg values associated with SST observations (Figure 10c) at the eastern coast of the Ligurian sea roughly correspond to the pattern observed for the RMSg associated with HFR data (Figure 10a). We believe the contribution of those SST observations to $\Delta$Tr is linked to that of HFR as their interaction is already observed in a similar 4D-Var application as reported by Bendoni et al. (2023)."

Lines 450-465 updated manuscript.

**4. Conclusion**

*76) 6L 429. « Corrections due to »*

We are referring to the corrections applied to the atmospheric forcing, not to the corrections to the model state due to atmospheric forcing.

**Figures and Tables**

All the modified figures requested by the reviewer are reported at the end of the present document indicating both the number they have in the draft and the number they have in the updated version of the manuscript (round brackets).

*77) Please remove « 2022 » from times axes in figures 2, 3, 6, 8 and 10, and put all the months.*
Done.

*78) Fig. 1a. The 2022 surface circulation of the FR is redundant with figure 5a (even if not at the surface). The 'real' known average circulation of 2022 should be shown for the reader to have a reference. You could use the 2022 mean surface (or 0-200m, maybe useful for Fig.5) circulation from the reanalysis of Escudier et al., (2021).*
We remove the average circulation from the panel a.

*79) Exchange 'b)' with 'c)', not the panel, only the letter.*
Done

*80) Here you provide general information about observations, so another issue here is linked with figure 3:*
*Fig. 3. Is the entire number of observation highly variable all along the 2022 year ?*
We added the time series of the total number of observations for each assimilation window over all the year as a subplot e) in Figure 1.

*81) With a look at all grey bars, it seems that the lack of data, that is present in a) and b) during January and July, is balanced by panel e) ?Please confirm or show me or in the paper the entire time series of number of data.*
See response to points 30 and 80 above.

*82) Fig. 4. W1 data is lower than 0,0 %. Can you provide a reason ? Overall, W1 data is not mentioned while being close to the ECC.*
The reason is that the relative amount of data is rounded to 0.0%. Indeed there is little data available for 2022 for platform W1. This is also the reason why we believe it does not significantly affect the ECC dynamic.

*83) Fig. 8d. Change FC to AC because you used FC for forecast (and adapt in the text).*
We changed it to AF (Atmospheric Forcing). We also changed it in Figure 11 and 12.

*84) Tab. 2. There no column for Nw. Can you provide a reason ? It is linked to Datum.*
There was an error, $N_w$ is the number assimilation window (see point 58 above).

*85) Fig. 9. Please label panels a-b « CoCM period (01 Jan–09 Apr) », panels c-d « HFR period (22 Apr-11 Aug) », and panels e-f « CoCM-HFR opposing period (18 Nov – 23 Dec) ». Please add in the caption « Periods are chosen with regard to Figure 8c » and adapt L 320-322.*
We added the labels as suggested and changed the caption accordingly.

*86) 'Sv*' is unclear. Please clarify whether it denotes Sverdrups divided by the horizontal area. If yes, use the unit 'Sv.m⁻²' and state that it represents depth-integrated transport per horizontal unit area, equivalent to the column-average velocity in the same units. If it is not area-normalized, please specify the denominator explicitly (e.g., 'per water column' or 'per grid point' instead of 'per grid cell'). For clarity and consistency, I recommend the area-normalized option or another explicit unit, used consistently across the text and figures. This clarification would strengthen the presentation.*

We modified the subplots as suggested by the reviewer, but we prefer to use Sv/km as we refer to the total transport across a section per unit width which is more suitable to evaluate or compare transports across sections. We also used the $\Delta F$ [Sv/km] instead of $\Delta Tr$ [Sv] in order to explicitly refer to a transport per unit width. We changed the caption accordingly and the equations in the text. See lines 372-373 updated manuscript.

*87) Fig. 12. I think that the caption is linked to an old figure.*
We corrected the caption.

**References**

*Please put the DOIs, which is specified in the guide for authors ([https://ocean-science.net/submission.html#references](https://ocean-science.net/submission.html#references))*
Done.

*The references' section needs data references to be accompanied by the notation [dataset] (author's guide), such as those in lines 24 to 28, and lines 147 to 156.*
We added [data set] in the references where needed.

*LI data from offshore the Gulf of Lion is be cited with Houpert but should be also cited as the way that the website asked at first.*
Done, we added Bosse et al. (2025).
- Bosse, A., Testor, P., Coppola, L., Bretel, P., Dausse, D., Durrieu de Madron, X., Houpert, L., Labaste, M., Legoff, H., Mortier, L., and D'ortenzio, F.: LION observatory data. SEANOE [data set], https://doi.org/https://doi.org/10.17882/44411, 2025.

*Cited references here in this review that are not in the preprint :*
*Napolitano et al., 2019, «The Tyrrhenian Intermediate Water...», PiO.*
*Iacono and Napolitano, 2020, « Aspects of the summer... », DSR P1.*
*McAdam et al., 2024, « Forecasting the Mediterranean... », SotP.*
*Estournel et al., 2025, «Extreme sensitivity... », Ocean Science.*
*Millot and Taupier-Letage, 2005, « Circulation in the Meditteranean Sea », book.*
*Garreau et al., 2018, « High-resolution Observations... », JGR Oceans.*
*Barral et al., 2025, « Assessment of the Water Mass Dynamics... », JGR Oceans.*
*Vara et al., 2019, « Role of atmosperic... », Ocean Modelling.*
*Doronzo et al., 2025, « Validating HF Radar... », Remote Sensing.*
*Fraysse et al., 2014, « Intrusion of Rhone River diluted ... », JGR Oceans.*
*Odic et al., 2022, « Sporadic wind-driven upwellin/downwelling... », CSR.*
*Vignudelli et al., 2004, « Distributions of dissolved... », ECS.*
*Lapucci et al., 2012, « Evaluation of empirical... », JARS.*
*Poulain et al., 2020, « On the dynamics in the ... », CSR.*

**Updated Figures list**

[Figure]

Figure 1 (Figure 1)

[Figure]

Figure 2 (Figure 2)

[Figure]

Figure 3 (Figure 3)

[Figure]

Figure 6 (Figure 5)

[Figure]

Figure 8 (Figure 6)

[Figure]

Figure 9 (Figure 8)

---

## Author Comment (AC2)

**Response to reviewer 2**
We thank the reviewer for the useful and inspiring comments. In the following, we first respond to the General observations and then we address point by point the Detailed comments. For the sake of clarity, we enumerated them. Line numbers in the response to comments refer to the revised version of the manuscript unless otherwise stated.

**General observations**

*The manuscript evaluates the impacts of different observation types on the Corsica Channel transport in a simulation of the NW Mediterranean Sea using the ROMS model with 4DVar data assimilation scheme. It follows a well-established methodology that have been applied many times to different regions around the globe. It represents an interesting contribution, that can be helpful in setting up a forecast system and to define future investments in the regional ocean observing system.*
*I feel the text would benefit from a major revision to strengthen its conclusions and improve the readability and figures. In particular, the discussion is fragmented and the figures are inconsistent in use of axes and colorbar ranges, making them hard to compare.*
*The conclusions summarize well the main results, but lack a closing statement that clearly defines the contribution of this paper and the expected importance in the larger picture.*
We thank the reviewer for her/his overall positive feedback and suggestions. We have revised the Introduction to improve its flow (see points 1, 3, 4, 5, 6, 7). We rearranged the structure of the Results and Discussion section splitting into two different sections: 3. Results and 4. Discussion. Section 3 is, in turn, split into 3.1 Performance of the DA system, and 3.2 Observation impact, whereas Discussion is composed of 4.1 Changes in circulation and 4.2 Insights into DA mechanisms (specifically see point 33 below). This makes the draft better structured and organized. We modified the colorbar of Figure 7 and Figure 10 (specifically see point 64 below) to make it more readable. We also added a closing statement in the Conclusion (specifically see point 45 below).
* * *
**Detailed comments**

**Introduction**

*1) Highlight the differences of your simulation to the Mediterranean Sea Physics Reanalysis, apart from resolution.*
Our setup differs from the Mediterranean Sea Physics Reanalysis (Escudier et al, 2021) in the following several aspects: a) horizontal resolution (see also point below); b) the use of 4D-Var method instead of 3D-Var; c) the assimilation of surface velocity from HF radar data; and d) the assimilation of in situ velocity data from the mooring system located in the Corsica Channel.
We modified part of the introduction as follows:
"In this paper, we present an improved version of the 4D-Var DA system previously implemented

by Bendoni et al. (2023) for the north-western Mediterranean, using the ROMS model (Shchepetkin and McWilliams, 2003, 2005). In addition to HFR-derived surface velocities and satellite SST, the system also assimilates SLA, in situ temperature, salinity and current profiles for the year 2022. It also represents an improvement over the Mediterranean Sea Physics and Reanalysis (Escudier et al., 2021), provided by the Copernicus Marine Environment Monitoring Service (CMEMS), by increasing the resolution from 1/24° to 1/36°, the use of a 4D-Var instead of a 3D-Var algorithm, and the assimilation of velocity observations".
Lines 83-89 updated manuscript.

*2) What difference does your higher resolution makes? Why is this necessary?*
The internal Rossby radius in the northwestern Mediterranean is of the same order of magnitude of the resolutions generally used in the literature (Beuvier et al., 2012). An increase in resolution is generally desirable to enrich the simulation and resolve smaller scales (i.e meso and submesoscale). We had to compromise between the resolution, the computational effort requested by the data assimilation procedure and the available computational power. Furthermore, with a view to employing the DA system as a donor model for downscaled coastal application, a step in resolution starting from 1/36° (our ROMS application) is preferable than 1/24° (NEMO).
- Beuvier, J., K. Béranger, C. Lebeaupin Brossier, S. Somot, F. Sevault, Y. Drillet, R. Bourdallé-Badie, N. Ferry, and F. Lyard (2012), Spreading of the Western Mediterranean Deep Water after winter 2005: Time scales and deep cyclone transport, J. Geophys. Res., 117, C07022, doi:10.1029/2011JC007679.

*3) Introduction lacks flow. In special the transition between the examples of 4DVar systems and the oceanography of the north-western Mediterranean Sea at lines 28-29.*
We reformulated the passage as follows:
 "...and in the Mediterranean Sea with specific application in the coastal area of the Tyrrhenian Sea (Iermano et al., 2016), in the Adriatic Sea (Janeković et al., 2020), and in north-western part of it (Bendoni et al., 2023).
The north-western Mediterranean Sea, also known as the Liguro-provençal Basin, is a crucial maritime…"
Lines 26-29 updated manuscript.

*4) Lines 29-35 – Will become more readable if broken up into smaller ones.*
We enlarged and rephrased this part of the introduction, now it reads:
"The north-western Mediterranean Sea, also known as Liguro-provençal Basin, is a crucial maritime region, hosting the Pelagos Sanctuary (Notarbartolo di Sciara et al., 2008) and several Marine Protected Areas (Francour et al., 2001). It is characterized by a cyclonic circulation (Northern Gyre) that involves Atlantic Water (AW) and modified Atlantic Water (mAW) at the surface, and Eastern Intermediate Water (EIW) and Tyrrhenian Intermediate Water (TIW) at intermediate depths. Winter Intermediate Water (WIW) is mostly located at intermediate depth in the western part of the basin, whereas Western Mediterranean Deep Water (WMDW) occupy the deeper layers (Astraldi et al., 1994; Napolitano et al., 2019; Schroeder et al., 2024; Barral et al., 2024).

Around Corsica, the Eastern Corsica Current (ECC) and the Western Corsica Current (WCC) flow along opposite coasts and converge to form the Northern Current (NC, or Liguro-Provençal-Catalan Current), which flows cyclonically along the Italian and France coasts, up to the Catalan coast (Astraldi et al., 1990; Millot, 1999). The ECC passes through the Corsica Channel (CC), a narrow strait between northern Corsica and the Capraia Island, about 30 km wide at the surface and 450 m deep. This strait represents the main connection between the warmer, saltier Tyrrhenian waters and the colder, fresher waters of the Liguro-Provençal basin (Bethoux, 1980; Astraldi et al., 1990).

The northward transport across the CC shows a marked seasonal cycle, with higher values during winter and spring and a net reduction in summer and autumn (Astraldi and Gasparini, 1992), occasionally reversing direction (Sciascia et al., 2019). This modulation aligns to the seasonal dynamics of the Tyrrhenian Sea. In winter, it is characterized by a large-scale cyclonic circulation, when both surface and intermediate waters flow along the Italian coast and bifurcate, one part reaching the CC, and the other veering southward to join a semi-permanent cyclonic structure close to the Bonifacio Strait called Bonifacio Gyre. In summer, most of the water mass is recirculated within the Tyrrhenian basin with little outflow toward the Ligurian Sea (Astraldi and Gasparini, 1994; Artale et al., 1994). In addition, the modulation of transport through the CC can be affected by the presence of a recurrent anticyclonic structure, peculiar of the summer season and located in the channel area, known as Ligurian Anticyclone (LA) (Ciuffardi et al., 2016; Iacono and Napolitano, 2020)."
Lines 29-50 updated manuscript.

*5) Lines 58-60 - This sentence is somewhat misleading, since modelling studies without DA can't by definition assess the impacts of DA. I recommend reviewing and potentially removing the reference to the non-DA simulations here.*

We reformulated the sentence as follows:

"Several modeling studies without DA (Béranger et al., 2005; Sciascia et al., 2019; Poulain et al., 2020) analyzed the dynamic of the CC. Those, instead, using DA (Vandenbulcke et al., 2017; Escudier et al., 2021) and including the CC in their computational domains lack a specific assessment of how assimilation improves the current representation in the channel."
Lines 73-75 updated manuscript.

*6) Lines 64-65 - I don't understand how the observation impacts can be used to evaluate the dynamical scales. Could you, please, explain.*

In this statement we mean that by identifying the spatial distribution of observations contributing to the variation of a certain quantity, it is possible to have an estimate of the spatial extent and scale involved. However, we agree with the reviewer that the statement is too generic and can be misleading. We modified it as follows:

"Evaluating the contribution of different observations to transport increments through the CC helps clarify how DA constrains the model through the relevant physical mechanisms. Furthermore, the spatial distribution of the assimilated observations to which the transport increment is most sensitive reveals the regions that are potentially most influential for transport variability."
Lines 79-82 updated manuscript.

*7) Lines 66-69 – It gives the impression that there are new observations assimilated that represent the "improvement" in relation to Bendoni et al. (2023). However, the first paragraph of session 2.1 gives a different idea. Therefore, this paragraph could use some rephrasing for more clarity.*

We modified the text at the beginning of Section 2.1

"The ROMS 4D-Var Data Assimilation system (Moore et al., 2011b) implemented for this study (version 4.3), builds upon the configuration developed by Bendoni et al. (2023), with modifications applied to the data assimilation framework, expanding the amount and type of assimilated observations (see Section 2.2) and by extending the analysis to a whole year."

Lines 97-99 updated manuscript.

**Session 2.1**

*8) Line 76 – Please specify the version of ROMS you used.*

Done, version 4.3, see also point 7 above.

*9) Line 111 – Can you justify your choice for the number of outer/inner loops and assimilation window length?*

The choice for the combination of inner/outer loops and assimilation window length is based on a compromise between literature values, and computational resources. Furthermore, since we focus on the observation impact methodology, and Levin et al. (2020) showed that the most part of the $\Delta$I is ascribable to the first outer loop, we opted for this choice.

We modified line 111 as follows:

"In this study, we use a single outer loop and 9 inner loops trying to find a compromise between the available computational power, the time required to run the experiments and the reduction of the cost function".

Lines 139-140 updated manuscript.

*10) Were there any sensitivity experiments?*

We performed some sensitivity experiments on the amount of inner loops, but not covering the whole year. Sensitivity experiments were principally performed on the type of assimilated observations and the characterization of the standard deviation for the calculation of the background error covariance matrix.

*11) Lines 112-114 – This interchangeable use of the terms "forecast" and "background" is confusing. Background is the accepted term used in data assimilation literature, while "forecast" is reserved to the "free run" initialized from an analysis. Please, review this and correct it through the text.*

We can understand the reviewer's request but in data assimilation the two terms can refer to the very same simulation and, based on the context, they have a different sense. In our framework, we use the same length for the forecast and for the assimilation window. We run the (n)th forecast for 3 days starting from the last time step of the previous (n-1)th analysis. Such a

forecast is indeed the background for the n(th) analysis cycle. When we refer, for example, to the run about which the nonlinear model is linearized, we use the term background. When, on the contrary, we refer to the skill and performance of the run starting from the analysis in an analysis-forecast sequence, we use the term forecast.

This is a case where background and forecast overlap, but they can differ (e.g. in case the assimilation window is 3 days and the subsequent forecast is 7 days). We believe it is important to keep them separated based on the context of reference.

*12) Moreover, I don't understand how this FR was run.*
The freerun FR was run starting from 2019 up to the whole 2022 and no assimilation was performed.

*13) Did you do a forecast after each analysis?*
Yes, we run the forecast after each analysis.

*14) Why not use your previous 4DVar analysis as initial conditions for the next assimilation window?*
It is what we did. The last time step of each analysis was used as the initial condition for the subsequent analysis window.

*15) Did you consider overlapping analyses?*
No, we did not consider overlapping analysis.

*16) Please, clarify how this was run and the rationale behind it.*
We reformulated the sentence as follows:
"The analysis/forecast sequence is characterized by a 3-day long assimilation window followed by a 3-day forecast, resulting in 122 windows over the year 2022. No overlapping between analysis (AN) is performed and each analysis uses the forecast (FC) from the previous window as background; hence, in the following, the terms "forecast" and "background" are used interchangeably. The choice of 2022 is motivated by the broader availability of HFR observations during this year."
Lines 140-144 updated manuscript.

**Session 2.2:**

*17) You give no explanation for how your observation errors were defined. The values look like what I would expect for instrument error, where representativeness errors would be expected.*
Observation errors are based on literature values and, in general, apart from the correct order of magnitude, they can be seen as a sort of tuning parameters. However, the calculation of superobservations (when several data are within the same computational cell), is performed as max(err_obs, std_data) where obs_err is the error assigned on the basis of literature values and std_data is the standard deviation of the data within the cell.

*18) In addition, there is no spatial structure to the errors. Could you, please, justify your choices?*

Spatial error is not included since the observation error covariance matrix R is assumed diagonal and does not contain spatial correlations among observations. This simplifies the assimilation procedure a lot. Attempts to relax this hypothesis are an active field of research:

- Goux, O., Weaver, A. T., Gürol, S., Guillet, O., & Piacentini, A. (2025). On the impact of observation‑error correlations in data assimilation, with application to along‑track altimeter data. Quarterly Journal of the Royal Meteorological Society, e5026.

*19) Lines 127-135 – The 3-hour low pass filter of the HFR velocities would still contain a tidal signal. However, you did not mention if and/or how you included tides in your simulation. Although the tidal signal is small in this area, this inconsistency can impact a "strong-constraint" DA scheme where all main physical processes are in principle included. Could you, please, elaborate how you deal with this?*

We did not take into account the tidal signal since tides in the area are in the order of centimeters and, furthermore, the boundary conditions are daily averaged values and do not contain the tidal signal. We did not perform a specific filtering procedure on the HFR velocities for simplicity and because we assumed the tidal signal in the velocity field would be negligible.

We added the following:

"Since the tidal signal in the area is of the order of centimeters, and considering that we use daily averaged values as boundary conditions, we did not take into account tides in the modelling system."

Lines 103-105 updated manuscript.

We also added the following in section 2.2 Assimilated observations:

"We did not perform a specific procedure to remove the tidal signal since we assumed the tidally induced velocities to be negligible."

Lines 161-162 update manuscript.

*20) Lines 145-168 – Please, break into separate paragraphs by observation type. This will help make it clear the different procedures applied to each observation.*

Done, it was an error to have a single paragraph.

*21) Line 146 – Please, add the reference for CORA.*

Done, we added the references:

- Szekely, T., Gourrion, J., Pouliquen, S., Reverdin, G., and Merceur, F.: CORA, coriolis ocean dataset for reanalysis. SEANOE [data set], https://doi.org/https://doi.org/10.17882/46219, 2025.

- Cabanes, C., Grouazel, A., von Schuckmann, K., Hamon, M., Turpin, V., Coatanoan, C., Paris, F., Guinehut, S., Boone, C., Ferry, N., de Boyer Montégut, C., Carval, T., Reverdin, G., Pouliquen, S., and Le Traon, P.-Y.: The CORA dataset: validation and diagnostics of in-situ ocean temperature and salinity measurements, Ocean Science, 9, 1–18, https://doi.org/https://doi.org/10.5194/os-9-1-2013, 2013.

*22) Lines 154-155 – I am curious why you chose to assimilate a SST product with a coarser resolution in relation to our model grid. Could you, please, justify your choice and the impact of errors that are carried from the SST product?*

There was a typo in the text and the correct product is the following product SST_MED_PHY_L3S_MY_010_042, at 1/20° resolution (that is still lower than our model resolution). The reference is corrected too:

- Pisano, A., Nardelli, B. B., Tronconi, C., and Santoleri, R. (2016). The new Mediterranean optimally interpolated pathfinder AVHRR SST Dataset (1982–2012). Remote Sensing of Environment, 176, 107-116. doi: https://doi.org/10.1016/j.rse.2016.01.019;

- Embury, O., Merchant, C.J., Good, S.A., Rayner, N.A., Høyer, J.L., Atkinson, C., Block, T., Alerskans, E., Pearson, K.J., Worsfold, M., McCarroll, N., Donlon, C., (2024). Satellite-based time-series of sea-surface temperature since 1980 for climate applications. Sci Data 11, 326. doi: https://doi.org/10.1038/s41597-024-03147-w.

We chose this product because, mostly focussing on the CC dynamics, we assumed that the 1/20° product was sufficiently resolved to constrain the transport across the channel. Furthermore, the use of a 1/100° product (the other available at a resolution higher than the model grid) would have required the super-observation procedure to reduce the resolution to 1/36° anyway. We did not fully understand what the reviewer meant with "the impact of errors that are carried from the SST product". Observation errors are taken into account with the **R** matrix.

*23) Lines 160-161 – Is the bias correction for the ADT comparable with the Reanalysis used for boundary conditions?*

The order of magnitude of the bias correction is comparable to the order of magnitude of the variability of the boundary condition water level from the Reanalysis.

*24) Lines 164-169 – Is the background QC applied to all the observations, or only the ADT? If only the ADT, why?*

Background QC was applied to all observation types. We used the same value for the beta parameter. Lines 196-200 updated manuscript.

**Section 2.3:**

*25) Lines 189-191 – It could be helpful to show the transect position in a map. Figure 1 is a good place to add this.*

We added the transect in Figure 1.

*26) Also, could you provide the rationale for choosing this particular section?*

The section corresponds to the Corsica Channel section employed in a previous study (Sciascia et al., 2019) and also to the location of the mooring system CoCM and dedicated cruise aimed at monitoring the dynamics of the corsica channel repeated CTD casts.

**Section 3.1**

*27) Lines 199-204 – Since you use the ratio between the eigenvalues as a guiding metric to identify overfitting, it would be better to show it in Figure 2.*
We modified Figure 2b as suggested.

*28) Line 205 – This is the first occurrence of "FR". Please, add the definition.*
We modified the sentence at line 91 as follows: "The nonlinear model without assimilation (freerun, FR) was run from 2019 to 2022, after…"
Line 116 updated manuscript.

*29) Line 217 – These should be positive values (a negative reduction would mean the simulations are going worse).*
We modified the text removing the minus sign. Line 251 updated manuscript.

*30) Line 225 – Is the difference in the fit of u and v related to the variability of the velocity components? It would be helpful to show the observation standard deviations next to the RMSEs.*
We believe that the difference in fit of u and v is principally due to the smaller magnitude of the u component (eastward) with respect to the v component (northward), that is also related to their variability. We think adding another source of information in the subfigures would produce too crowded plots. However, we mention the possible explanation for the difference in the fit between u and v in the discussion subsection 4.1.
"If we focus on the physical processes affecting the performance of the DA system we can try to explain, for example, why the error reduction is more pronounced for the v than the u velocity component observed by the CoCM (Figure 4a, b, e and Table 1). Since the latter has a lower observed average magnitude (0.008 m/s) and standard deviation (0.022 m/s) with respect to the former (0.05 m/s and 0.09 m/s, respectively), the DA has more chances to correct v than u, also because we made the assumption that the background error covariance is proportional to the standard deviation of the state variables. being the flow orographically constrained, the eastward velocity (u) cannot reach large values and possible small variations are more hardly caught by the model."
Lines 398-404 updated manuscript.

*31) Lines 237-241 – Please, present the standard deviations for all observations to support your affirmation. This could be added to Table 1.*
Done, updated Table 1.

*32) Lines 251-256 – All that the time series show is data was in fact assimilated.*
We agree with the reviewer and we further speculated in point 36 below.

**Session 3.2**

*33) There is a lot of jumping back and forth with the figures, which harms the readability of the paper. Please, consider rearranging your discussion.*

The reviewer is right and we rearranged the Results and Discussion section by splitting it into two separate sections: 3 Results and 4 Discussion. They are further divided into 3.1 Performance of the DA system, and 3.2 Observation impact. The Discussion is composed of 4.1 Changes in circulation and 4.2 Insights into DA mechanisms.

*34) Lines 260-262 – I disagree that that graphs show that there is no BIAS. A better way to show this is by plotting the bias in Figure 3. Moreover, the graph for in situ S does not look "evenly distributed" for innovations.*

We understand the point raised by the reviewer. However, we stated there is not a "significant bias", not that the bias is absent. As mentioned above, we think that adding the bias or other quantities to the subplots of Figures 3 and 4 would make them too crowded. We reformulated the text as follows, also considering that for salinity the reviewer is right.

"Overall, the points in Figure 7 are approximately evenly distributed between positive and negative innovations for all observation types, except for in situ salinity (Figure 7f). This can be viewed as an indication that a clear and evident bias is not present in these observed components of the state vector, but for the in situ salinity for which a non symmetric distribution for positive and negative innovation values is present (44.4% negative **d**; 55.7% positive **d**)."
Lines 309-312 updated manuscript.

*35) Lines 273-276 – Please explain how can the impacts on transport shown in Figure 7 be many times higher than the values and variability shown in Figure 8.*

The top left of each subplot of Figure 7 reports the scale value by which the transport value has to be multiplied to get the actual values.

*36) Lines 276-281 – So, the observational studies consider the velocity constant with longitude? And how do you explain the difference to the other model studies? It is not good enough to say additional analysis could be done. Please, explain this better.*

Yes, in general observational studies consider the northward velocity constant with longitude because data are available only at the vertical line of the mooring. The point is that the FR is in line with other model-based studies, whereas the discrepancies come out after the assimilation is performed. We believe that such a difference can be ascribed to the assimilation procedure and until we perform DA over other years, it is difficult to understand if 2022 was a particular year of low transport or not.
We modified the text as follows:
"Recalling that the yearly averaged transport values that we obtained at the CC for FR, FC and AN are equal to 0.49, 0.31 and 0.28 Sv, respectively, these values are lower than previous estimates based on observations, ranging from 0.54 Sv to 0.71 Sv (Astraldi and Gasparini, 1992; Astraldi et al., 1994), and numerical models, such as 0.49 Sv in Sciascia et al. (2019) and 0.5 Sv in Béranger et al. (2005), the latter based on a multi-decadal time series. However, reanalysis by de la Vara et al. (2019) and Barral et al. (2024) obtained an average transport around 0.33-0.34 Sv and 0.35 ±0.365 Sv, respectively, considering the whole transect including the CC and the area from Capraia Island to the Italian coast.

Part of the discrepancy with observational estimates may stem from the assumption of negligible longitudinal variability in the northward velocity across the channel that is not fully supported by our model results (Figure 5e and Figure 8d, e) and was also observed by previous numerical studies (Sciascia et al., 2019). Interestingly, the two reanalyses of de La Vara et al. (2019) and Barral et al. (2024) obtained an annual average transport comparable to our result by considering the whole Tyrrhenian transect. These values lower than those from previous literature can be ascribed to a possible negative annual averaged transport for the area between the Island of Capraia and the Italian Coast, or to a more appropriate representation of the flow in the area.

The transport across the whole transect linking the Tyrrhenian and Ligurian Seas, and the interaction between the two sub-transects, also in light of basin-scale mass budgets, deserve further studies and analysis. Furthermore, additional analysis, particularly over a longer time period, will help determine whether the year 2022 reflects inter-annual variability (Vignudelli et al., 1999), or is the signal of a longer-term trend."

Lines 338-349 updated manuscript.

*37) Lines 282-287 – What explains this variability and the differences in the assimilation model?*

We did not fully understand the point raised by the reviewer. The variability between winter and summer season is a characteristic of the CC transport. We believe that the difference between DA-informed simulations and freerun, especially during the summer season, is mainly due to the mismatch between observations from the CoCM and the freerun during such a period.

*38) Lines 304-308 – Please, explain why the impact per observation is bigger in situ T and S in relation to HFR.*

We believe that in situ T and S data have a larger impact per observation since they directly affect the whole water column, whereas HFR data influence the surface layer and can have only an indirect effect on the water column.

We added the following:

"This can be explained by the fact that in situ data directly affect the whole water column, whereas surface data can only influence it indirectly, through the adjoint."

Lines 332-333 updated manuscript.

*39) Lines 309-313 – This calls for a look at the vertical structure (transect) of the increments. You could link to the discussion of Figure 9.*

We reorganized the paragraphs: first we talk about the overall reduction in transport in Figure 5 (Figure 6 in the original manuscript), then we analyze the different trends of the increment based on the period of the year (Figure 6; Figure 8 in the original manuscript) and analyze their structures (Figure 8; Figure 9 of original manuscript). Finally, we link the comments to Figure 8a to the yearly-averaged velocity distribution of the increment δv.

Now the paragraphs read:

"Comparing the non-assimilative (FR) and the assimilative (FC and AN) runs it is important to stress that the latter are the cumulative result of corrections applied to the background by DA over each assimilation window. Observing Figure 5, it is clear that DA systematically reduces the northward velocity in the channel to better match the observations, leading to a corresponding reduction in total transport (Figure 6a). However, Figure 6b, shows that northward transport reduction is dominant up to mid-August 2022, after which positive transport

increments become more frequent than negative ones. This evolving pattern is further dissected in Figure 6c: from January to early April, CoCM data primarily drive the transport corrections; between April and early August, HFR data become more influential, despite not being directly located at the CC transect, where the transport is computed. Figures 8a and c show the average velocity increment δv during the periods dominated by the JERICO facilities, CoCM (from 01 January to 09 April), and HFR data (from 22 April to 11 August), respectively. CoCM observations tightly constrain the flow near their location but with the side effect of accelerating the northward flow in the surface western part of the transect. In fact, the δv values, averaged over the whole year, are characterized by an overall reduction for the most part of the CC transect and a slight increase at the upper western flank (not shown). This may be the result of insufficient corrections to the boundary conditions, redirecting the volume transport to unconstrained portions of the domain (Figure 8a, Figure 5e). Interestingly, it can help to explain why the total transport (Figure 6a) is less affected than the velocity alone (Figure 5a, b, c) when the FR and the AN runs are compared. Figures 5d and e show that while the northward flow weakens across most of the transect in AN with respect to FR, particularly on the eastern side (that observed by the mooring), a partial compensation occurs through a weaker (in absolute value) southward flow on the western flank and the already mentioned slightly northward acceleration at the upper western flank."

Lines 354-371 updated manuscript.

Here, we report the yearly averaged value of δv.

[Figure]

40) Lines 339-347 – You kind of hint at the potential issue here: The placement of your southern boundary. Given the importance, please expand your discussion.

We added the following:

"Additional analyses might expand the computational domain to include an additional portion of the domain which potentially influences the transport through the CC. However, it is not straightforward to identify the extension which maximizes the impact of initial condition and atmospheric forcing at the expense of boundary condition. Furthermore, extending the domain to contain the whole Western Mediterranean basin would excessively increase computational costs."

Lines 393-397 updated manuscript.

See also the discussion at point 67 from Reviewer 1.

*41) Line 348 – Here is a jump in your discussion. You should add some explanation and refer to session 2.3.*

After the rearrangement of the Results and Discussion section, the statement to which the reviewer is referring is now moved at the beginning of subsection 4.2 Insight into DA mechanisms and it reads:

"Observation impact can allow us to investigate in which ways DA modifies the model state with respect to a particular metric (transport across a section in our case) by optimally balancing between observations and background solution."

Lines 431-433 updated manuscript.

We also added the following:

"Developing more the analysis, we can estimate the contribution of each observations to the transport increment through $\Delta Tr_{TL} = \mathbf{d}^T \cdot \mathbf{g}$. The presence of a seasonality in $\Delta Tr$ when we consider observation typology (Figure 6c), that is roughly cold months controlled by CoCM while warm months until late autumn by HFR, could be explained by the corresponding seasonality in the CC transport. Small northward velocity values in warm months tend to produce small innovations d and, consequently, possible small $\mathbf{d}^T \cdot \mathbf{g}$. To better control the dynamic at the channel, DA relies on another source of observations that is HFR. In cold months, the opposite occurs and the role of CoCM observations is predominant.

Moreover, the variability of the elements of $\mathbf{g}$ reveals the location of observations which have the greatest influence on the model. Figure 10 shows the spatial distribution of the root mean squared values of the elements of g (RMSg) calculated over each computational cell for each observation type."

Lines 439-447 updated manuscript.

*42) Lines 352-355 – This again shows the weight given to your southern boundary, on top of potential issues on the way the river flux is added to your simulation. How do the sea surface salinity maps look?*

Salinity surface maps show plumes at the mouth of Rhone and Arno rivers.

Here the yearly averaged surface salinity and standard deviation maps for the Analysis run are reported.

[Figure]

Please, consider also the response to point 72 from Reviewer 1.

*43) Are the river plumes well represented?*

From the qualitative point of view, by graphically inspecting the modelled sea surface salinity, the plumes are well represented. However, we did not specifically calculate the model skills with respect to observations close to the mouths, as this was not the primary objective of the work.

Instead, we focused on the whole behavior of the salinity field, whose representation is improved by the assimilation procedure compared to the FR.

*44) Lines 380-395 – This is a very interesting analysis! As the heat flux reflects only what is happening at the surface, it would be good to see what is happening below. I believe the water column heat content or vertically integrated density would be helpful here.*

We thank the reviewer for this request since it helped us to notice that our interpretation of the above-mentioned process had some flaws. ROMS is a Boussinesq model, and for the Boussinesq approximation the density variations are neglected in the momentum equations, except in the buoyancy term along the vertical coordinate, and the continuity equation reduces to conservation of volume. As a consequence, steric expansion cannot be directly modelled.

For these reasons, we reformulated the whole paragraph trying to interpret the results in the light of the above-mentioned aspects.

We modified the lines 380-403 as follows (we also modified Figures 11 and 12):

"For most assimilation windows the primary way DA constrains the CC transport is the modification of the boundary conditions (Figure 6d). By selecting those with the largest positive and negative transport increments largely attributable to BC modifications (90-th and 10-th percentiles of the $\Delta Tr_{BC}$ distribution, respectively), we calculated the northward velocity increment at the southern boundary $\delta v$ (Figure 11a and b), the free surface height increment $\delta \eta$ (Figure 11c and d), and net heat flux increment $\delta Q_{net}$ (Figure 11e and f), between analysis and background, averaged over the two sets of assimilation windows. A positive $\Delta Tr_{BC}$ is indeed associated with a northward-inducing $\delta \eta$ (Figure 11c) that is consistent with a clear northward barotropic acceleration on the Tyrrhenian side and a less intense southward barotropic acceleration, distributed over a much wider cross sectional area, at the Ligurian-Provençal basin side (Figure 11a). For the negative $\Delta Tr_{BC}$ such a mechanism is more marked for both $\delta v$ and $\delta \eta$. The barotropic nature of these increments is confirmed by the almost uniform increment velocity distribution along depth (Figure 11a and b).

Interestingly, the $\delta Q_{net}$ is characterized by a pronounced surface heating/cooling in the Ligurian-Provençal basin when the CC transport increases/decreases (Figure 11e and f). Since the ROMS model adopts the Boussinesq approximation, such a mechanism cannot be related to steric adjustments, but rather to a modification to the baroclinic pressure field to maintain the consistency between the barotropic correction at the boundaries and observations in the interior. Additional analysis is required to make explanatory hypotheses at the base of the observed mechanism.

Astraldi and Gasparini (1992) proposed that CC dynamics are governed by atmospheric conditions over the western part of the Ligurian-Provençal basin, where winter heat loss enhances the steric gradient between the Tyrrhenian and Ligurian-Provençal basins, driving Tyrrhenian waters into the Ligurian Sea and strengthening the ECC. Our results cannot directly mimic the processes described by Astraldi and Gasparini (1992), however, focusing now on the largest positive and negative transport increments attributable solely to the correction to atmospheric forcing, $\Delta Tr_{FC}$ (95-th and 5-th percentiles of the $\Delta Tr_{AF}$ distribution, respectively), we observe that $\delta \eta$ (Figure 12a and b) and $\delta Q_{net}$ (Figure 12c and d) align with the mechanism proposed by Astraldi and Gasparini (1992): an increase (decrease) in transport is triggered by a differential surface cooling (warming) with associated eastward (westward) positive gradient in

the free surface increment. Even in this case, additional analyses are required to delve into the mechanism responsible for the agreement with literature results."

Lines 484-507 updated manuscript.

Here, we also report the updated Figure 11

[Figure]

And updated Figure 12

As requested by the Reviewer, we report below the relative increment in density δρ/ρ associated to the analysis reported in Figure 12, averaged over the upper 50 meters and along a longitudinal transect at 42.5° Latitude. It is clear that the cooling/heating produces increase/decrease in density mainly at the surface. However, this is not associated with a shrinking/expansion of the related sigma layers. For this reason we reported in the above-mentioned modified paragraph: "such a mechanism cannot be related to steric adjustments, but rather to a modification to the baroclinic pressure field to maintain the consistency between the barotropic correction at the boundaries and observations in the interior."

[Figure]

**Session 4**

*45) I feel there is a missed opportunity here to discuss how effective the different observations are in constraining the simulation and the potential implications for forecasting and the "greater" western Mediterranean Sea – the usual goal of an assimilative system. You have lots of information that can be helpful in driving the evolution of the observing system.*
*A closing paragraph would do the job.*
In the conclusion we added the following paragraph.
"This work represents an additional step toward an operational 4D-Var data assimilation forecasting system for the Northwestern Mediterranean sea, given the significant effect the employed observations have in constraining the circulation of the area for both the analysis and the forecast stages."
Lines 538-540 updated manuscript.

**Tables**

We understand the point of the reviewer but we try to better explain the reasons for our setting.
In general in the Data Assimilation community, a reduction in error is an improvement (so the negative values mean improvement), whereas an increment in correlation is an improvement as well (so the positive values mean improvement). We are saying that if you reduce the error or increase the correlation, you have a good result. For bias, we want it to be reduced in absolute value, so the negative values mean the model is improved.
The absolute values for the RMSE are already reported in the Figures and we would like to be more informative by explicitly showing how much the assimilation procedure can improve the simulations.

We modified the caption of Table 2 as follows:
"Root mean square impact for the different observation typologies, both globally (Global refers to the effect on transport of all observations belonging to a certain typology), and considering the average contribution of a single observation (Datum refers to the average effect on transport of a single observation belonging to a certain typology). The last column reports the total number of observations per type."

**Figures**
All the modified figures requested by the Reviewer are reported at the end of the present document indicating both the number they have in the draft and the number they have in the updated version of the manuscript (round brackets).

We added a contour line to the HFR area and we removed the streamlines since information about the yearly averaged flow field is reported in Figure 9. In this way the figure is more readable.

We added the section in the subplot a).

*50) b) I feel the total observation density to not be very informative, and would be better plotted as a "pcolor" or "contourf". Instead, the distribution of each observation type can give a better idea of the coverage and expected impacts.*

Subplot b) (c in the update version of the manuscript) has been produced as a pcolor coloring each cell of the model domain based on the amount of observations within it. The information about the distribution of each observation typology is partially contained in Figure 10.

*51) c) Why do you have an empty depth bin close to 2000m?*

Because accidentally no data are available for that depth interval.

*52) Figure 2 – Please, show the ratio between eigenvalues in sublot (b).*

We modified Figure 2 as requested.

*53) Figure 3 – Could you, please, use the same axis ranges for the number of observations in all your subplots? This will make it easier to compare.*

We understand the request of the reviewer, but we think it would lead to an excessively flattened representation for in situ data with respect to HFR and SST data.

*54) Figure 4 – As for figure 3, please use the same axis limits for the number of observations.*

Please, see the previous response for point 53 above.

*55) It would also be interesting to have the std of the (u,v) observations to be able to evaluate how much the variation of the fit by depth is relative to the variability of the currents.*

As already mentioned in a previous comment, we prefer to avoid putting the std of the data in the plot and we added such information in Table 1. We discussed the effect of current variability in model skill at the point 30 above.

*56) Figure 5 – The reference to the subplots in your label is wrong. Always add the unit after the depth.*

We corrected the references in the caption ('c' instead of 'd', and 'd' instead of 'e'). We added the depth unit both in the caption and in the text in the subplots. (This is Figure 9 in the updated manuscript).

*57) Figure 6- Please, add the units for depth.*

Done, both in the caption and in the text within the subplots. (This is Figure 5 in the updated manuscript).

*58) Figure 7 – Please, have use the same axis and colorbar for all your subplots.*

Using the same colorbar range and axis would lead to a non-readable figure. The x-axis can have values in the range -0.25/0.25 or -2.5/2.5; y-axis are multiplied by $10^{-4}$ or $10^{-3}$ and the colorbars have maximum values which differ by almost 2 degrees of magnitude. We instead changed the colormap to increase the readability.

*59) This figure is underexplored in the text, which makes me question if it is necessary.*

The use of this type of figure is a common practice in data assimilation papers dealing with observation impact (report references) and it gives you information about the correctness of the procedure. We refer to this in the text at lines 304-322 (updated manuscript).

*60) Figure 8 – b) This is unnecessary. You don't seem to explore this figure enough in your text.*
Subplot b, by showing the equivalence between the transport increment calculated from the nonlinear model and that obtained by the observation impact procedure, ensures the tangent linear assumption holds for the adopted framework (duration of assimilation window, typology if metric employed). We are reluctant to the proposed change as we think it is important to provide this information.

*61) Subplots c) and d) arguably contain the most important results in your manuscript, and should be deeper explored in your text. Try to keep in mind what is the main point you want to make with your paper.*
We agree with the reviewer about the importance of subplots c and d.
After the rearrangement of the structure of the paper, the comments associated with these two subplots gained a larger exposure and are not hidden by the surrounding text.

*62) Figure 9 – Please, add your section position to the maps.*
The section has been added to the plots in green. (This is Figure 8 in the updated manuscript).

*63) Figure 10 – Please, use the same limits for your map colorbars.*
The use of the same limits for all the colorbars would produce a loss in information content since the different colorbars span two orders of magnitude.

*64) The colormap you used is not helpful either. More contrast between values would make it easier to spot differences.*
We agree with the reviewer and the colormap has been changed.

**Updated Figures list**

[Figure]

Figure 1 (Figure 1)

[Figure]

Figure 2 (Figure 2)

[Figure]

Figure 5 (Figure 9)

[Figure]

Figure 6 (Figure 5)

[Figure]

Figure 7 (Figure 7)

[Figure]

Figure 9 (Figure 8)

[Figure]

Figure 10 (Figure 10)

[Figure]

Figure 11 (Figure 11)

[Figure]

Figure 12 (Figure 12)